# RAS-inhibiting biologics identify and probe druggable pockets including an SII-α3 allosteric site

Katarzyna Z. Haza [1,3], Heather L. Martin [1,3], Ajinkya Rao [1,3], Amy L. Turner [1,3], Sophie E. Saunders [1], Britta Petersen[1], Christian Tiede [1], Kevin Tipping[1], Anna A. Tang [1], Modupe Ajayi[1], Thomas Taylor[1], Maia Harvey[1], Keri M. Fishwick [1], Thomas L. Adams [1], Thembaninkosi G. Gaule[1], Chi H. Trinh [1], Matthew Johnson[2], Alexander L. Breeze [1], Thomas A. Edwards [1], Michael J. McPherson[1] & Darren C. Tomlinson [1 ✉]

RAS mutations are the most common oncogenic drivers across human cancers, but there remains a paucity of clinically-validated pharmacological inhibitors of RAS, as druggable pockets have proven difficult to identify. Here, we identify two RAS-binding Affimer proteins, K3 and K6, that inhibit nucleotide exchange and downstream signaling pathways with distinct isoform and mutant profiles. Affimer K6 binds in the SI/SII pocket, whilst Affimer K3 is a non-covalent inhibitor of the SII region that reveals a conformer of wild-type RAS with a large, druggable SII/α3 pocket. Competitive NanoBRET between the RAS-binding Affimers and known RAS binding small-molecules demonstrates the potential to use Affimers as tools to identify pharmacophores. This work highlights the potential of using biologics with small interface surfaces to select unseen, druggable conformations in conjunction with pharmacophore identification for hard-to-drug proteins.

[1] School of Molecular and Cellular Biology, Astbury Centre for Structural and Molecular Biology, University of Leeds, Leeds, UK. [2] Avacta Life Sciences, Wetherby, UK. [3] These authors contributed equally: Katarzyna Z Haza, Heather L Martin, Ajinkya Rao, Amy L Turner. ✉email: d.c.tomlinson@leeds.ac.uk

The RAS family of small GTPases consists of four members, KRAS4A, KRAS4B, HRAS, and NRAS, which act as bi-directional molecular switches that cycle between an inactive GDP-bound form, and an active GTP-bound form[1]. Mutations in RAS are the most common oncogenic drivers, with KRAS being the most frequently affected member; especially in pancreatic, lung, and colon cancer[1]. This makes RAS a strong therapeutic target, but despite having been identified as a drug target for over 30 years, only recently have compounds been developed that show promise in pre-clinical trials[2]. This paucity of agents has been caused by the lack of clearly druggable pockets on the surface of RAS. However, recent work has identified two pockets that may be amenable for drug binding[3–11]. The first of these, the SI/II-pocket, exists between the Switch I and Switch II regions of RAS in an area involved in the binding of the nucleotide exchange factor, Son of Sevenless (SOS). Several groups have independently developed compounds that bind this pocket with varying affinities and efficacies, predominantly in the micromolar range[5–8], except for BI-2852 which has nanomolar binding affinity and efficacy[11]. The second, the SII-pocket, is located under the Switch II loop and was identified using a tethered warhead approach relying on the reactive nature of the cysteine in the G12C mutant[9,10]. This pocket is not fully formed in published KRAS structures in the absence of the inhibitor; however, a groove is evident in some structures and it has been identified as a potential allosteric site computationally[3,4]. The development of tethered compounds targeting this pocket has led to the only series of RAS inhibitors currently in clinical trials[12,13]. This compound series is limited to cancers harboring G12C mutations, however, a cyclic peptide, KRpep-2d with a preference for G12D mutations has been identified that binds in a similar pocket[14,15] showing that biologics can also probe pockets in KRAS. It would be interesting to determine whether this pocket can be non-covalently exploited in other RAS isoforms and mutants, giving wider application to RAS-driven cancers.

Targeting of RAS has also been explored using scaffold-based biologics, besides the cyclic peptide KRpep-2d. Antibodies and their alternatives have been developed that bind RAS with nanomolar affinities, inhibiting nucleotide exchange, interactions with RAF, and activation of downstream pathways concurrent with negative impacts on RAS-induced cell growth and transformation[16–21]. Amongst others, these include scFvs, DARPins, and monobodies[16,17,19,21,22]. Although to date some of these have been used to assist the identification of small molecules[5,7,23], none have directly probed druggable pockets on RAS, as the majority of these scaffold-based biologics tend to bind over large protein interfaces[16,17,19,22] that are difficult to mimic with small molecules[24]. As some scaffold-based biologics form smaller interfaces[21,25,26] there emerges the tantalizing prospect that biologics could be used as tools to identify druggable pockets and novel conformers, and could also have the potential to act as pharmacophore templates for in silico-informed drug discovery. Here, we explore the possibility of using Affimer proteins, an established biologic with a small probe surface formed by two variable regions[26] known to bind at protein interaction 'hotspots'[25,27,28], to probe RAS for druggable pockets and conformers that might be amenable to small-molecule inhibition. Such a direct approach utilizing small probe surfaces has not previously been used with scaffold-based biologics and could revolutionize drug discovery, by exemplifying a pipeline for small molecule design that has the potential to unlock the currently 'undruggable' proteins.

Here, we demonstrate the use of Affimer proteins, a scaffold-based biologic, to identify and directly probe two druggable pockets on wild-type KRAS associated with inhibition of nucleotide exchange and effector molecule binding. The Affimer that binds to the SI/SII pocket actually mimics the current small molecule inhibitors that target the pocket as seen in a competitive NanoBRET assay, providing a proof-of-principle for using Affimer-target interfaces as pharmacophore templates. The Affimer that binds the SII region selects a previously unseen conformer of the pocket present in wild-type KRAS. The Switch II region adopts a more open position, demonstrating that selecting and targeting this site via non-covalent binding is possible. Our work supports two important concepts in the use of biologics: firstly, they can be used to select for, and stabilize, conformations that are only present as a small fraction of the conformations of the target protein in solution, particularly those that may not be present in extant crystal structures; and secondly, scaffold-based biologics can act as drug discovery tools and/or pharmacophore templates for identification and development of small-molecule inhibitors. This approach is likely to be applicable to other important therapeutic targets and presents the exciting potential for an alternative pipeline for drug discovery.

## Results

**Identification and biochemical characterization of anti-RAS Affimers.** Seven unique Affimer proteins that bind wild-type KRAS in both the inactive GDP-bound form and the active form, bound to the non-hydrolyzable GTP analog-GppNHp, were isolated by phage display[29] (Supplementary Table 1). To identify inhibitors of RAS, these Affimer proteins were screened at 10 μM for their ability to inhibit SOS1-mediated nucleotide exchange, the primary process in RAS activation. Three of the Affimers, K3, K6, and K37, showed clear inhibition of this process (Fig. 1a), the remaining four Affimers that bound to KRAS showed partial (Affimers K19 and K68) or no inhibition (Affimers K69 and K91) of nucleotide exchange. The latter four Affimers are not discussed further, besides K69 which was used as a control in the NanoBRET assays. The dose-dependency of nucleotide exchange inhibition by the 3 selected Affimers was then assessed (Fig. 1b). Affimer K3 displayed the greatest inhibition of nucleotide exchange on wild-type KRAS with an $IC_{50}$ of $144 \pm 94$ nM, with Affimers K6 and K37 also displaying strong inhibition with $IC_{50s}$ of $592 \pm 271$ nM and $697 \pm 158$ nM, respectively (Supplementary Table 2). Next, the abilities of the inhibitory RAS-binding Affimers, K3, K6, and K37 to disrupt the interaction of RAS with its effector protein RAF were determined by KRAS:RAF immuno-precipitation experiments using purified proteins (Fig. 1c and d). Again, all three Affimer proteins caused a significant reduction in the amount of KRAS immunoprecipitated, with K3 being the most potent with a 79% reduction compared to a control Affimer in which the variable regions are AAAA and AAE, respectively, while K6 and K37 showed 40% reductions ($p < 0.0001$, $p = 0.0307$ and $p = 0.0439$, respectively; One-way ANOVA with Dunnett's post hoc test).

Next, having been selected against KRAS, the specificity of the Affimer proteins for distinct RAS isoforms was assessed by nucleotide exchange assays. Whilst K6 and K37 showed no isoform specificity (Supplementary Table 2), Affimer K3 demonstrated a degree of isoform specificity, with weaker inhibition of HRAS and no measurable inhibition of NRAS with $IC_{50}$ values of $144 \pm 94$ nM for KRAS, $2585 \pm 335$ nM for HRAS and not obtainable for NRAS, respectively. The effects of the Affimer proteins on mutant forms of KRAS were also evaluated by nucleotide exchange assays with recombinant G12D, G12V, and Q61H KRAS mutants. Only Affimer K3 displayed a distinct mutant profile with 20-fold weaker inhibition of Q61H ($IC_{50} = 3005 \pm 865$ nM), suggesting specificity towards wild-type KRAS and G12 mutations.

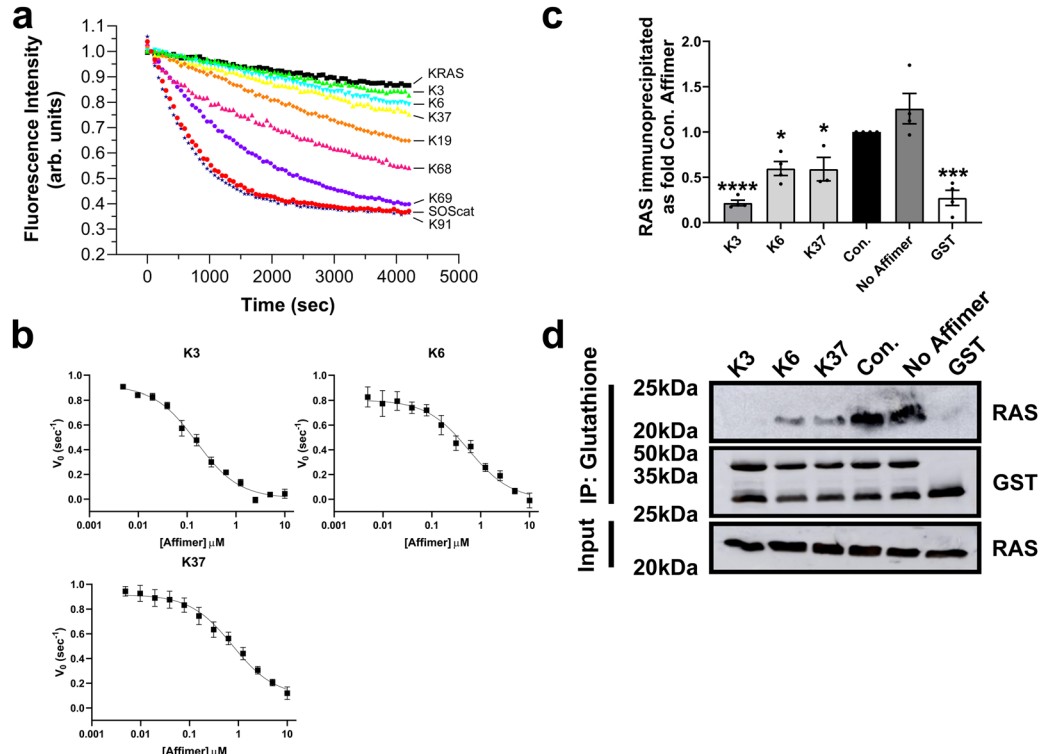

**Fig. 1 Biochemical analysis of RAS-binding Affimers. a** Nucleotide exchange assay shows 3 Affimers, K3 (green triangles), K6 (turquoise triangles), and K37 (yellow triangles), inhibit SOS1-mediated nucleotide exchange, whilst K19 (orange diamonds) and K68 (magenta triangles) show inhibition of intrinsic nucleotide exchange and K69 (purple hexagons) and K91 (navy stars) do not inhibit nucleotide exchange at 10 μM. KRAS alone is shown as black squares and in the presence of SOS1cat as red circles. **b** Affimers K3, K6, and K37 demonstrate dose-response inhibition of KRAS$^{WT}$ nucleotide exchange (Data fitted to the Hill Model ([Affimer] vs. response (three parameters)), $n = 3$ independent experiments for K3 and K6 and $n = 5$ for K37. **c** Immunoprecipitation of KRAS with GST-RAF-RBD is inhibited by the RAS-binding Affimers, K3, K6, and K37 compared to control Affimer (Variable regions of AAAA and AAE) which does not differ from the no Affimer (PBS). GST alone does not pull down RAS (Quantification using ImageQuantTL; $n = 3$ independent experiments). **d** A representative Western blot of the pull-down experiment from **c**. Data are mean ± SEM, One-way ANOVA with Dunnett's post-hoc test *$p = 0.0307$ K6, $p = 0.0439$ K37, ***$p = 0.0002$ GST, ****$p < 0.0001$ K3. SOScat catalytic domain of SOS1 (Son of Sevenless). Con. Control Affimer, RBD RAS Binding Domain, IP immunoprecipitation, GST Glutathione-S-Transferase.

**Affimer proteins bind to intracellular RAS and inhibit downstream signaling.** We then examined whether the Affimer proteins retained their ability to interact with, and inhibit RAS in human cells by using HEK293 cells and transiently transfecting in plasmids for expression of His-tagged Affimer proteins. Affimers K3, K6, and K37 all showed the ability to pull down endogenous RAS, while the control Affimer showed no such activity (Fig. 2a), demonstrating binding within live cells. To understand the effects of Affimer protein binding to endogenous RAS on downstream signaling, activation of the MAPK pathway was explored. HEK293 cells were transiently co-transfected with plasmids encoding turbo-GFP (tGFP)-tagged Affimers and FLAG-tagged ERK1 expressing constructs and were then stimulated with epidermal growth factor (EGF)—this stimulation normally induces phosphorylation of ERK1 via MAPK. A co-transfection approach was used to ensure assessment of ERK1 phosphorylation in Affimer expressing cells only, a similar approach has been previously used[20]. All three Affimer proteins significantly reduced phosphorylation of the recombinant ERK1 (One-way ANOVA with Dunnett's post-hoc test $p = 0.0002$ K3, $p < 0.0001$ K6, and $p < 0.0001$ K37, respectively). However, the effect of Affimer K3 was lower in magnitude than those of K6 and K37, with K3 showing only a 31% reduction compared with an 85% reduction for K6, and a 69% reduction for K37 (Fig. 2b and c).

To further study the impacts of RAS-binding Affimer proteins on ERK phosphorylation, we developed an immunofluorescence assay to allow the phosphorylation levels and nuclear translocation of endogenous ERK to be examined. HEK293 cells were transiently transfected with tGFP-tagged Affimer-expressing constructs, stimulated with EGF, fixed and stained with an anti-phospho-ERK (pERK) antibody, and analyzed for alterations to nuclear pERK levels. In accordance with results from the immunoprecipitation experiments, expression of all three Affimer proteins resulted in significant reductions in pERK levels (One-way ANOVA with Dunnett's post-hoc test $p < 0.0001$ for all three Affimers), with K3 having a less pronounced effect (ca. 50% reduction for K3, compared with K6, 90% and K37, 85% reductions) (Fig. 2d and e). There were no significant differences ($p = 0.429$ One-way ANOVA), in the percentage of tGFP positive cells between the different Affimer constructs. We determined if the decrease in pERK nuclear translocation correlated with increased Affimer expression on a cellular level by determining pERK inhibition at a number of different tGFP intensities. Increased tGFP expression, and thus Affimer expression, showed a reduction in pERK nuclear translocation with a plateau reached at 1000 arbitrary units (Fig. 2f). We speculated that the lower level of inhibition observed in K3-transfected cells was due to the RAS variant specificity of this Affimer: HEK293 cells express all RAS genes, but predominantly HRAS[30], against which K3 is 20-fold less active. To test this hypothesis, the Affimer proteins were transfected into Mouse Embryonic Fibroblasts (MEFs) that have been engineered to express single human RAS genes (gift from William Burgen at Fredrick National Laboratory for Cancer Research, Maryland, USA). All three Affimer proteins showed

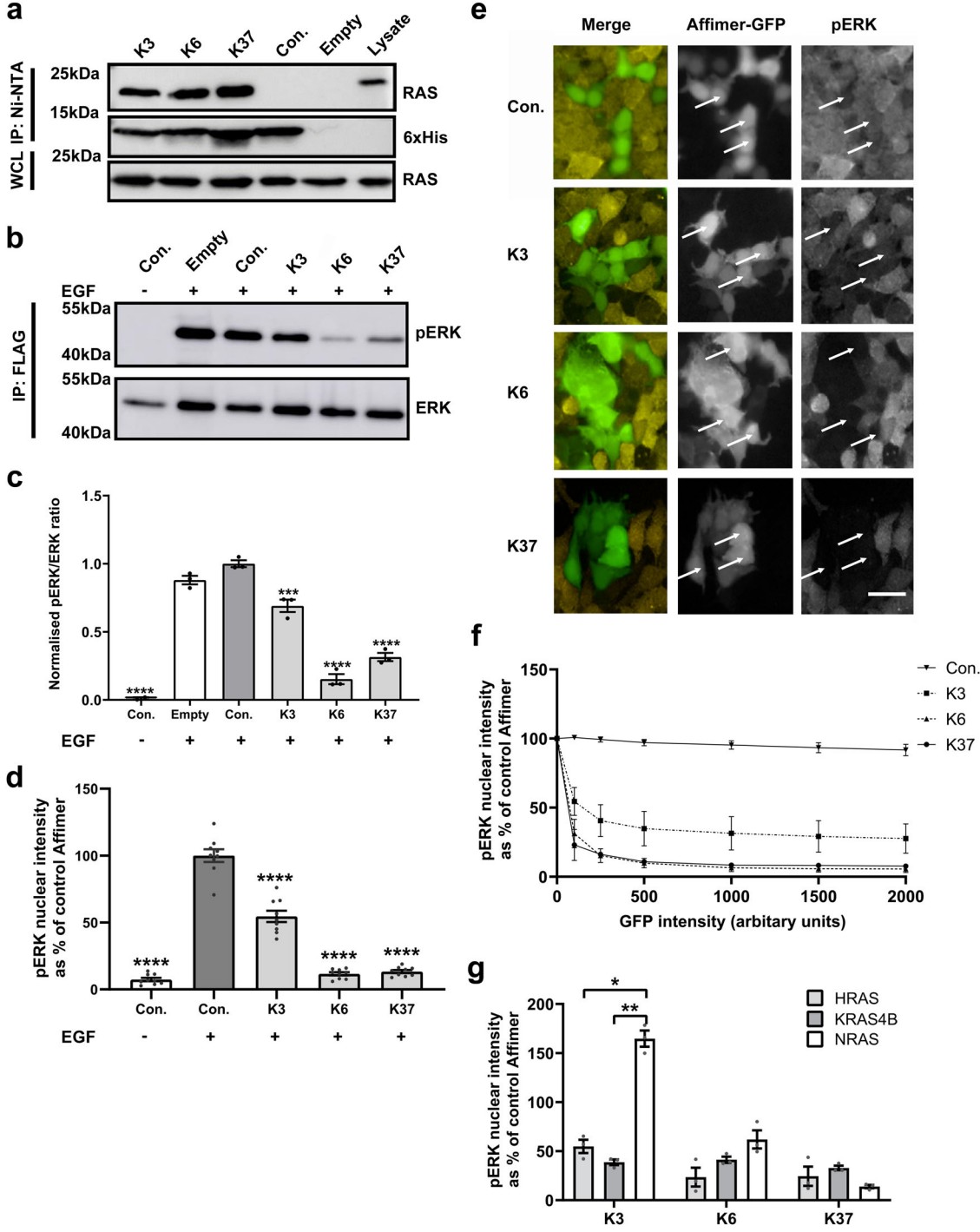

robust inhibition of ERK phosphorylation in KRAS-expressing MEFs with K6 and K37 showing a similar level of inhibition in MEFs expressing HRAS and NRAS. By contrast, K3 showed variant specificity, with inhibition of NRAS significantly ablated whilst the degree of inhibition for HRAS was reduced, but not significantly when compared with that of KRAS (Two-way ANOVA with Tukey's post hoc test $p = 0.0237$ for NRAS vs. HRAS and $p = 0.0096$ for NRAS vs. KRAS) (Fig. 2g). Thus, the cellular data support the biochemical data that Affimer K3 has a preference for KRAS over NRAS, with HRAS values being intermediate.

To evaluate the impact of the Affimer proteins on MAPK signaling in the presence of mutant, oncogenic forms of KRAS, the following cancer cell lines were utilized: Panc 10.05

(KRAS$^{G12D}$), SW620 (KRAS$^{G12V}$), and NCI-H460 (KRAS$^{Q61H}$). As anticipated from the biochemical data, all three Affimer proteins showed robust inhibition of FLAG-ERK1 phosphorylation in Panc 10.05 and SW620 cells ($p < 0.0001$ All Affimers in Panc 10.05 cells and $p = 0.0026$ K3, $p = 0.0054$ K6 and $p = 0.0088$ K37 in SW620 cells, respectively, One-way ANOVA with Dunnett's post-hoc test) (Fig. 3a and b). However, in the Q61H mutant NCI-H460 cells despite all three Affimers showing inhibition of FLAG-ERK1 phosphorylation, the magnitude was reduced by 30–40% for K3 and K37 compared to the other mutant cell lines ($p = 0.0269$ K3, $p = 0.0062$ K6, $p = 0.0210$ K37, One-way ANOVA with Dunnett's post hoc test) (Fig. 3c) supporting the biochemical data that the Affimer proteins show mutant specificities. It is possible that the differences in responses to

**Fig. 2 Affimers bind to intracellular RAS and inhibit downstream signaling. a** Ni-NTA immunoprecipitation of transiently expressed intracellularly His-tagged Affimers with endogenous RAS from HEK293 cells. RAS-binding Affimers, K3, K6, and K37, pulled down endogenously expressed RAS, whilst the control Affimer did not. A representative blot from 3 independent experiments is shown. **b, c** HEK293 cells were co-transfected with FLAG-ERK1 plasmid and pCMV6 encoding tGFP tagged Affimers. Twenty-four hours post-transfection, cells were serum-starved and treated with EGF for 5 min. FLAG-ERK1 was precipitated from cell lysates and analyzed for phosphorylation. **b** shows a representative blot from 3 independent experiments quantified in (**c**) showing that Affimers K6 and K37 significantly reduced ERK phosphorylation by over 60% while Affimer K3 reduced it by 30%. (One-way ANOVA with Dunnett's post-hoc test **$p = 0.0002$ K3, ****$p < 0.0001$ K6, and $p < 0.0001$ K37). **d** RAS-binding Affimers reduce EGF-induced phosphorylation and nuclear translocation of endogenous ERK in HEK293 cells as measured by immunofluorescence as a percentage of the control Affimer, with Affimers K6 and K37 showing inhibition of over 80% whilst Affimer K3 inhibit by 50% in GFP-expressing cells over 1500 arbitrary units (One-way ANOVA with Dunnett's post-hoc test ****$p < 0.0001$, $n = 3$ independent experiments). **e** Representative images of the effects of RAS-binding Affimers, K3, K6, and K37, and the control Affimer on EGF-stimulated upregulation of pERK in HEK293 cells. A selection of GFP-positive cells (green) expressing RAS Affimers (arrowed) show reduced staining for pERK (yellow). Scale bars are 50μm. **f** Assessment of Affimer expression level on pERK inhibition as determined by immunofluorescence, increased GFP expression, and thus Affimer expression shows a reduction in pERK nuclear translocation ($n = 3$ independent experiments). **g** RAS-binding Affimers inhibition of EGF-induced phosphorylation and nuclear translocation of endogenous ERK in mouse embryonic fibroblasts (MEFs) expressing single human RAS isoforms as measured by immunofluorescence as a percentage of the control Affimer. Affimers K6 and K37 shown inhibition in all RAS isoforms, whilst Affimer K3 inhibited KRAS and HRAS to a lesser degree with no inhibition of NRAS (Two-way ANOVA with Tukey's post-hoc test *$p = 0.0237$ HRAS vs. NRAS and **$p = 0.0096$ KRAS vs. NRAS, $n = 3$ independent experiments). Data are mean ± SEM. Con. Control Affimer (Variable regions of AAAA and AAE), EGF epidermal growth factor, tGFP turbo green fluorescent protein, WCL whole cell lysate, IP immunoprecipitation, Empty transfection reagents only.

Affimers between cell lines are a result of variations in Affimer-expression levels relative to RAS. The impacts of the Affimer proteins on mutant KRAS were therefore also tested using an immunofluorescence assay in conjunction with MEF cells expressing KRAS mutants; G12D, G12V, and Q61R allowing only cells with high Affimer-expression where it is probable that RAS is saturated to be analyzed. Only Affimer K3 in the Q61R background showed a significant lack of inhibition of pERK nuclear intensity ($p = 0.0221$ Two-way ANOVA with Tukey's post-hoc test) (Fig. 3d), supporting the observation that Affimer K3 distinguishes between KRAS Q61 mutants and other KRAS variants. Together, these cellular data confirm that RAS-binding Affimer proteins are functional in cells, reducing ERK phosphorylation and that Affimer K3 demonstrates RAS variant and mutant specificity in cells, as well as in vitro. Given the similarities in biochemical profile and cellular activities between K6 and K37, together with a similar amino acid sequence motif (Supplementary Table 1), we postulate that they are likely to bind the same epitope although K37 showed less potency in the nucleotide exchange assays. We, therefore, focused on Affimer proteins K3 and K6 in our subsequent structural studies.

**K6 binds the SI/II hydrophobic pocket on KRAS.** The crystal structure of Affimer K6 in complex with GDP-bound wild-type KRAS was determined at 1.9 Å resolution, revealing that Affimer K6 binds to a shallow hydrophobic pocket on KRAS between the switch regions (Fig. 4a and Supplementary Table 3). The Affimer K6 binding site overlaps that of SOS1 providing structural evidence that K6 acts as a SOS1 competitive inhibitor (Supplementary Fig. 1a). The binding site further overlaps with that of the RAS-binding domain (RBD) of RAS-bound RAF (Supplementary Fig. 1b) supporting the RAS:RAF immunoprecipitation results. Affimer residues 40–45 from variable region 1 are important in the binding interface between KRAS and Affimer K6 (Fig. 4b and c). The Affimer tripeptide motif formed of P42, W43, and F44 binds the shallow hydrophobic pocket of KRAS (Fig. 4b–top left panel). The P42 residue forms no interactions with KRAS, however, it is critical for function suggesting it has a geometric and structural role facilitating the interactions formed by W43 and F44. W43 of Affimer K6 and V7/L56 from KRAS form a hydrophobic cluster strengthened by Affimer residues T41 and Q45 forming hydrogen bonds with KRAS switch region residues D38/S39 and Y71, respectively (Fig. 4b–top right panel). The importance of these amino acid residues for K6 function was

confirmed by mutational analysis. Individual replacement of P42, W43, F44, or Q45 with alanine reduced Affimer-mediated inhibition of nucleotide exchange (Fig. 4d). These data also revealed the importance of residues F40, N47, and R73 for the inhibitory function of Affimer K6. Indeed, complete removal of the second variable region abolished the inhibitory ability of K6 (Fig. 4d ΔVR2). This effect is most likely a result of F40, N47, and R73, and Q45 forming intra-Affimer hydrogen bonds that stabilize the tripeptide, P42, W43, F44 (Fig. 4b–bottom panel). These data suggest that the functional Affimer motif responsible for binding and inhibition of KRAS is small, in agreement with the total interacting interface estimated by PISA analysis[31] of 478.3 Å² for the K6:KRAS complex, a substantially smaller area than most common protein-protein interaction surfaces. The combination of a functional motif and small interaction interface provides a strong basis from which to consider the development of small molecule inhibitors.

Affimer K6 binds the SI/SII pocket that has been previously documented[3–8,11], and alanine scanning has suggested a functional pharmacophore from K6 is responsible for the observed inhibition. This PWFQxN peptide motif is also present in Affimer K37 and it is thus likely that K37 interacts in a similar manner to K6, but this remains to be confirmed. This binding pocket has previously been identified, and a number of small molecules exist to target it, including DCAI[6], compound 13[8], Abd-7[7], and BI-2852[11] (Supplementary Table 3). We measured the affinity of K6 for both GDP-bound and GppNHp-bound KRAS by SPR to test whether this was comparable to these small molecules. Affimer K6 bound both forms of KRAS with low nanomolar affinities and showed a significant preference for GDP-bound KRAS ($K_D = 1.36 \pm 0.87$ nM for GDP and $K_D = 7.88 \pm 1.09$ nM for GppNHp, Student $t$-test $p = 0.0095$) (Fig. 5a and b). Thus, Affimer K6 has a 10-fold higher affinity for KRAS than Abd-7[7], the strongest-binding small molecule with a $K_D$ of 51 nM (c.f 750 nM for BI-2852[11], 1.1 mM for DCAI[6] and 340 μM for compound 13[8], see Supplementary Table 3). We also inspected the SI/SII-binding small molecules for structural similarities to the K6 pharmacophore (Fig. 4e). All of the small molecules have an aromatic ring that inserts into the pocket and which is reproduced by the side chain of W43 of Affimer K6. However, only K6 and BI-2852 appear to interact across the whole of the pocket surface. This suggests that additional points of interaction may underlie efficacy, as BI-2852 is the most potent of the compounds to date[6–8,11]. Affimer K6 shows a similar degree of potency to

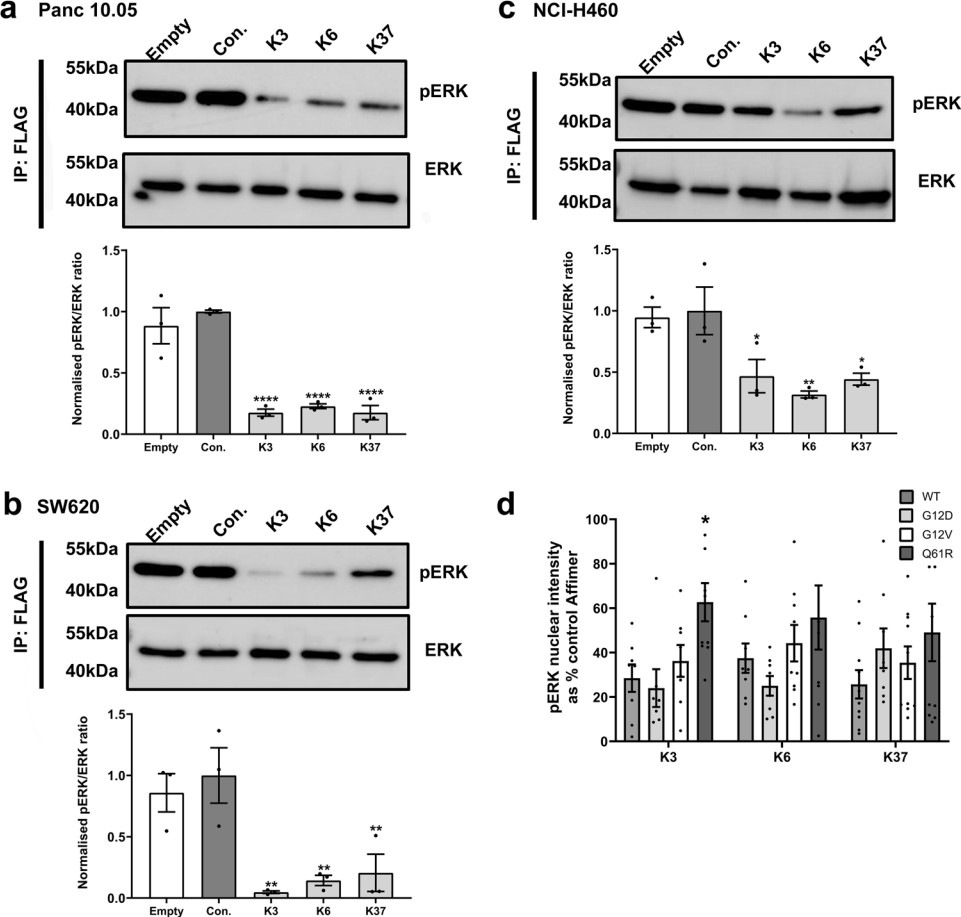

**Fig. 3 RAS-binding Affimers show different mutant specificities. a** Panc 10.05 (KRAS^G12D), (**b**) SW620 (KRAS^G12V), and (**c**) NCI-H460 (KRAS^Q61H) cells were co-transfected with FLAG-ERK1 plasmid and pCMV6 encoding tGFP tagged Affimers. Twenty-four hours post transfection cells were serum-starved for 1 h. FLAG-ERK1 was precipitated from cell lysates using anti-FLAG beads and analyzed for phosphorylation by immunoblotting with anti-ERK and anti-phospho-ERK antibodies. Representative blots are shown together with quantification graphs. All three RAS-binding Affimers, K3, K6, and K37, inhibit ERK phosphorylation in Panc 10.05 cells (**a**) One-way ANOVA with Dunnett's post-hoc test compared to control Affimer, ****$p < 0.0001$) and SW620 cells (**b**) One-way ANOVA with Dunnett's post-hoc test compared to control Affimer **$p = 0.0026$ K3, $p = 0.0054$ K6, and $p = 0.0088$ K37). The magnitude of inhibition by Affimers K3 and K37 is reduced in NCI-H460 cells (**c**) One-way ANOVA with Dunnett's post-hoc test compared to control Affimer *$p = 0.0269$ K3, **$p = 0.0062$ K6, *$p = 0.0210$ K37). **d** RAS-binding Affimers inhibit EGF-induced phosphorylation and nuclear translocation of endogenous ERK in mouse embryonic fibroblasts (MEFs) expressing single human KRAS mutants (G12D, G12V, Q61R) as measured by immunofluorescence as a percentage of the control Affimer. Only Affimer K3 shows weaker inhibition in the Q61R expressing cell line (Two-way ANOVA with Tukey's post-hoc compared to KRAS^WT for each Affimer *$p = 0.0221$. Data are mean ± SEM, $n = 3$ independent experiments for all cell lines. Con. Control Affimer (Variable regions of AAAA and AAE), EGF epidermal growth factor, tGFP turbo green fluorescent protein, IP immunoprecipitation, Empty transfection reagents only, WT wild type.

BI-2852, both showing IC$_{50}$ values in the nanomolar range for inhibition of nucleotide exchange (IC$_{50}$ = 592 ± 271 nM for Affimer K6 and IC$_{50}$ = 490 nM for BI-2852[11]).

**K3 locks wild-type KRAS in a conformation harboring a SII/α3 pocket**. Affimer K3 lacks the PWFQxN motif of Affimers K6 and K37 and has a different biochemical profile in terms of mutant and isoform specificities. The underlying reasons for this were confirmed by determining the crystal structure of Affimer K3 in complex with the GDP-bound form of KRAS to 2.1 Å resolution (Fig. 6a and Supplementary Table 3), however, crystallization of this complex was difficult leading to lower than anticipated Rfactor and FreeRfactor values (see "Methods" section for details). Affimer K3 binds KRAS with high affinity irrespective of the nucleotide bound ($K_D = 59.4 ± 15$ nM for GDP-bound and $K_D = 44.4 ± 0.8$ nM for GppNHp-bound) (Fig. 5c and d).

Variable region 1 binds KRAS between SII and the α3 helix, with residues 41–46 being crucial for interaction (Fig. 6b and c) and inhibition. Indeed, the functional importance of these Affimer residues was highlighted by mutational analysis, as individual alanine substitutions abolished the inhibition of SOS1-mediated nucleotide exchange (Fig. 6d). Affimer K3 residue D42 bridges the gap between SII and α3 helix, forming hydrogen bonds with R68 and Q99, respectively (Fig. 6b–top left panel). The KRAS:K3 binding is strengthened by Affimer K3 residue D46, binding both Q99 and R102 of the α3 helix (Fig. 6b–top left panel), as well as Y45 of K3, forming a hydrogen bond with E62 of SII (Fig. 6b–top right panel). This forms a larger binding surface allowing Affimer K3 residues I41 and I43 to form hydrophobic interactions with V103 and M72, and V9 of KRAS, respectively (Fig. 6b–bottom left panel). In addition, the backbone carbonyl of W44 forms a hydrogen bond with H95 of the α3 helix, as well as orientating its indole side chain towards KRAS so that it is packed against the

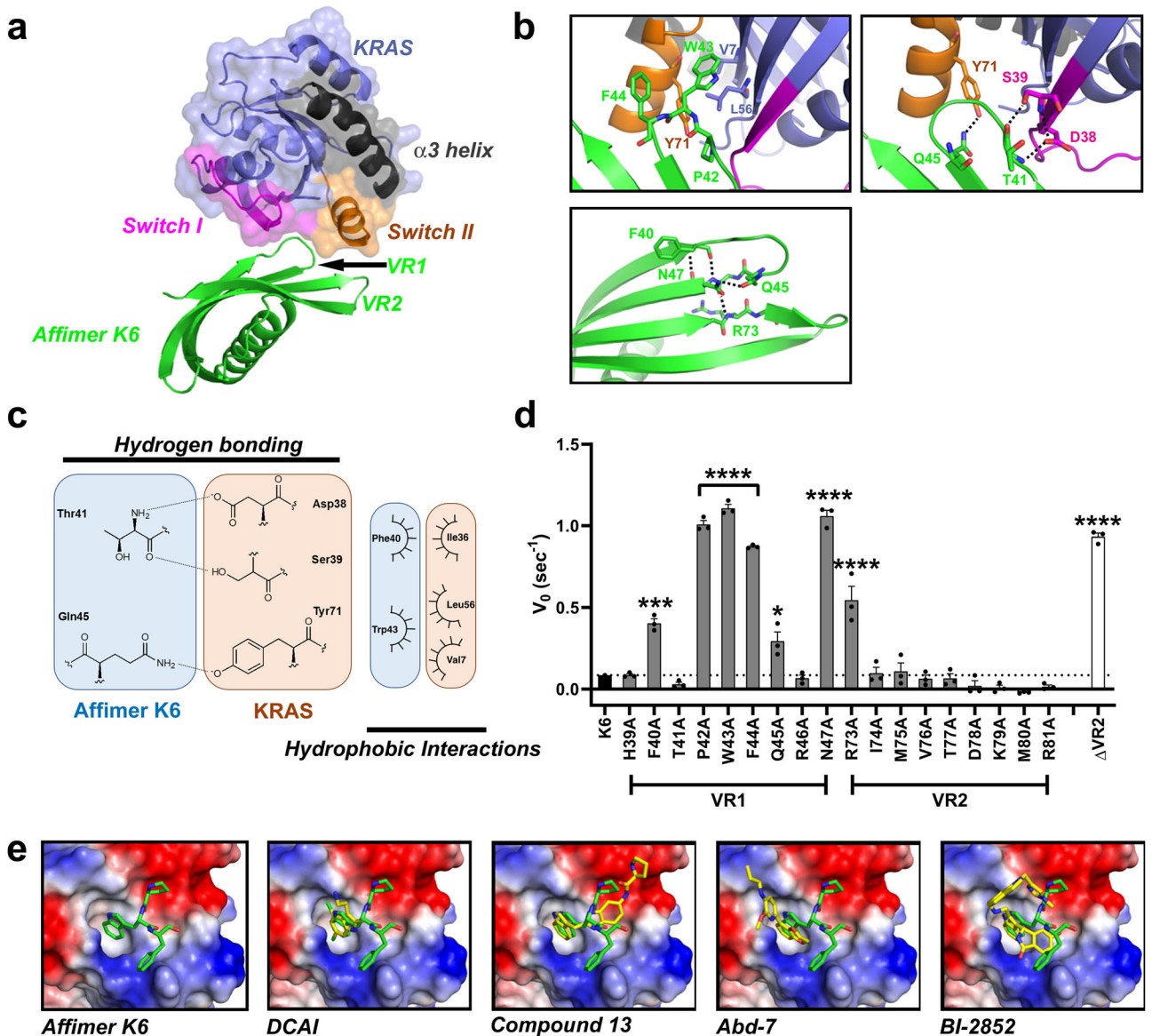

**Fig. 4 Variable region 1 of Affimer K6 binds between Switch I and Switch II of KRAS. a** Affimer K6 (green) was co-crystallized with KRAS$^{GDP}$ (slate) and solved to a resolution of 1.9 Å. The switch I (magenta), switch II (orange) and α3 helix (black) are depicted, showing their relative positioning around variable region 1 of Affimer K6. **b** Intramolecular and intermolecular interactions in the KRAS:Affimer K6 co-crystal structure are depicted; black dotted lines represent the hydrogen bonds that stabilize the critical hydrophobic contacts. **c** Affimer K6 (VR1) and KRAS interactions shown in planar form. Hydrogen bonds are shown as black, dotted lines between the contributing atoms; additional hydrophobic interactions are represented by arcs, their spokes radiating towards the residues they contact. (Data was generated using PDBePISA (CCP4i)[31] and verified in MacPyMOL). **d** Alanine scanning data of the variable regions of Affimer K6 highlights Affimer residues important for inhibition of nucleotide exchange and the importance of VR2. This highlights the residues that are important for both KRAS:K6 interactions and intra-Affimer interactions that stabilize the conformation of Affimer K6. Unaltered K6 is shown in black, variable region 1 residues are shown in dark gray, variable region 2 residues in light gray, and removal of variable region 2 (ΔVR2) in white. ($n = 3$ independent experiments). **e** Comparison of Affimer K6 tripeptide, P42, W43, F44, (green) with the small molecules (yellow) that bind the same SI/SII pocket. Data are mean ± SEM, One-way ANOVA with Dunnett's post hoc test *$p = 0.0224$ Q45A **$p = 0.0002$ F40A ****$p < 0.0001$ P42A, W43A, F44A, R73A, and ΔVR2. Images were generated in MacPyMOL v1.7.2.3, and ChemDraw Prime 16.0. VR variable region.

residues of Q61, H95, and Y96 (Fig. 6b–top right panel). The involvement of Affimer K3 residue W44 in binding H95 may explain the specificity of K3 for KRAS seen in the biochemical and cellular data, as H95 is a unique residue only present in KRAS and not in HRAS and NRAS. The importance of H95 in mediating Affimer K3 selectivity for KRAS was confirmed by immunoprecipitation assays of mutants H95Q and H95L that mimic the corresponding residues in HRAS and NRAS. Mutating H95 affected the ability of Affimer K3 to immunoprecipitate

KRAS, with mutation to glutamine (as found in HRAS) reducing the amount of RAS pulled down by 45% and mutation to a hydrophobic leucine residue (as found in NRAS) abolishing RAS immunoprecipitation completely, thus giving support to our previous observations (Fig. 6e).

The inter-molecular interactions described above cause significant conformational shifts in the effector lobe of KRAS, most notably, the α2 helix of SII being forced further from the α3 helix, as compared to the WT-KRAS$^{GDP}$ crystal structure (PDB

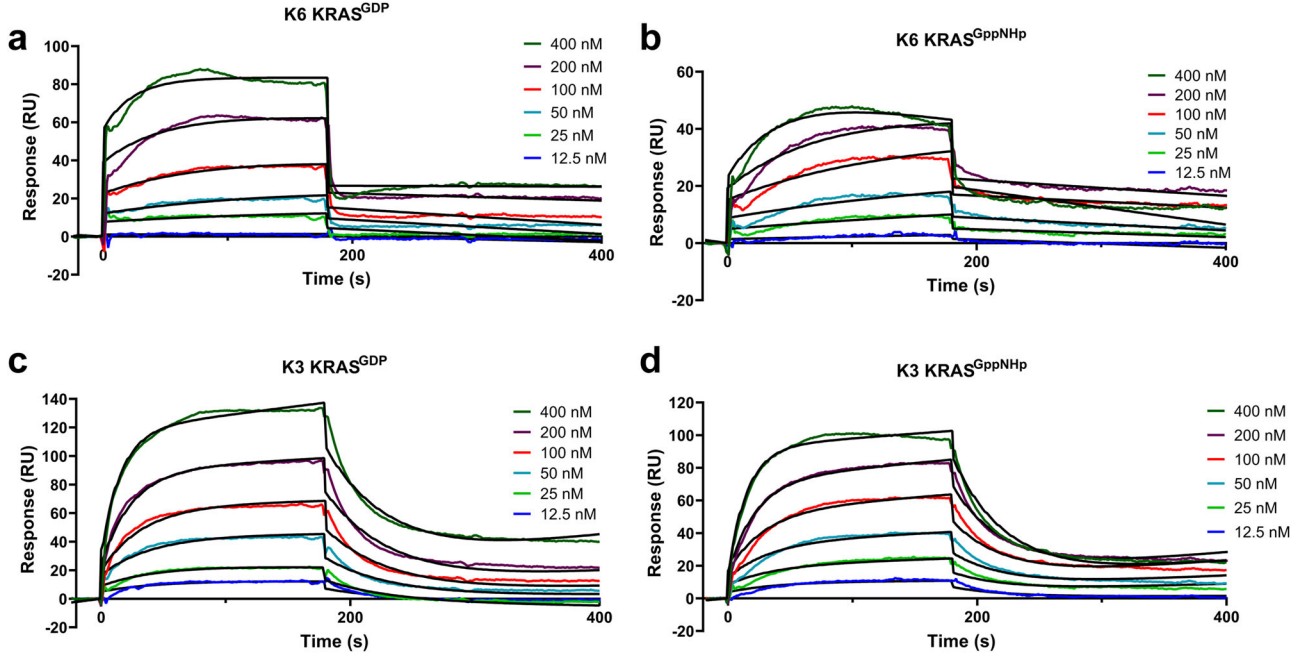

**Fig. 5 Affimers K3 and K6 bind KRAS with nanomolar affinity.** SPR measured binding activities for Affimer K6 with GDP bound KRAS (**a**), GppNHp bound KRAS (**b**) and Affimer K3 with GDP bound KRAS (**c**), GppNHp bound KRAS (**d**). Affimers were immobilized on streptavidin-coated CM5 sensor chips via C-terminal biotin and differing concentrations of GDP or GppNHp bound KRAS flowed over. Representative curves of 3 replicate experiments are shown with experimental data in color and Langmuir 1:1 fitting curves in black.

code: 4OBE[32]) (Fig. 6f). The observed conformational shift originates from the flexible glycine residue (G60) of the DxxGQ motif of SII. The N-terminal loop of SII is further rearranged, orienting itself over the K3 binding motif towards the α3 helix. These changes in SII conformation not only generate a larger binding surface but also facilitate a number of intramolecular hydrogen bonds in KRAS, distinct to WT-KRAS$^{GDP}$ (PDB code: 4OBE[32]). We postulate that the binding of Affimer K3 residue D42 to KRAS R68, by a salt bridge interaction, shifts the R68 residue into an orientation necessary to facilitate a hydrogen-bonding network between E37 of SI, A59, G60, S65, and R68 of SII (Fig. 6b–bottom right panel), thereby stapling the SII region to the SI site. This hydrogen bonding network is not seen in WT-KRAS$^{GDP}$ (PDB Code: 4OBE[32]), or WT-KRAS$^{GppNHp}$ (PDB code: 6GOD[5]). It is possible these global conformational shifts, together with specific KRAS residue changes, such as E37, and even M67 whose side chain abrogates the RAF-RBD interface may explain the ability of K3 to abolish the KRAS-RAF interaction (Supplementary Fig. 1d).

Furthermore, as SII acts as the main anchor point for SOS1 binding[33,34], the significantly reduced flexibility of this site may, together with occlusion of the Cdc25 domain of SOS1 forming a steric clash (Supplementary Fig. 1c), underlie the K3-mediated inhibition of SOS1-mediated nucleotide exchange. We postulate that K3 binding locks KRAS in an inactive conformation by stapling the switch regions together through induced hydrogen bonding, reducing conformational dynamics required for its activity.

The dynamic freedoms of SII are further reduced by the folding of the N-terminal loop of SII over the K3 binding motif resulting in a hydrogen bond interaction between Q61 of SII and Y96 of the α3 helix. This locks Q61 in a position distal to the active site[33,35]. This involvement of Q61 supports the biochemical and cellular data, showing the loss of inhibition of nucleotide exchange and a reduction in EGF-stimulated pERK nuclear translocation in NCI-H460 cells and Q61R

MEFs when Q61 is replaced with a histidine or arginine residue (KRAS $^{Q61H/Q61R}$).

Thus, Affimer K3 has identified a conformer of wild-type KRAS that generates a druggable pocket, with an estimated interface area of 790.6 Å (PISA (EMBL-EBI)[31] analysis), buried between the SII region and the α3 helix, not previously observed in wild-type KRAS. A similar pocket has previously been reported in the KRAS$^{G12D}$ mutant in a complex with a cyclic peptide, KRpep-2d[15]. However, there are critical differences between the pocket bound to K3 and when bound to KRpep-2d. Notably, K3 binding involves the KRAS-specific residue H95 giving rise to the KRAS-specificity seen in our cellular assays, the isoform specificity of KRpep-2d has not been assessed to our knowledge. In addition, K3 binding induces the KRAS intra-molecular bonds between Q61 and Y96 without the involvement of residue 12, whereas KRpep-2d requires an aspartic acid residue to coordinate the same intramolecular bonding network[15]. Further-more, a comparable groove has also been documented in the KRAS$^{G12C}$ mutant, together with a small molecule series that inhibits RAS function via binding at this pocket[9,10,12] (Fig. 6g). Of this series, the most recently published AMG510 compound has reached clinical development[12]. The compound is covalently tethered to the C12 residue and explores the same SII/α3 helix pocket. We hypothesize that AMG510 induces a similar mode of inhibition seen by K3, whereby the switch regions are stabilized by the ligand. However, it is clear that although the AMG510 compound and Affimer K3 explore the same cryptic groove, the conformations of SII lead to distinct pocket conformations with widely different electrostatics. Affimer K3 stabilizes a more open conformation compared to the closed conformation seen with AMG510 (Fig. 6g). Further to this, binding of K3 to KRAS decreases flexibility by inducing hydrogen bond interactions between SI/SII, and SII/α3 helix, which is not present in the KRAS$^{G12C}$:AMG510 structure (PDB code: 6OIM[12]). The shape, size, and physiochemical composition of the pocket identified by Affimer K3 suggests a potentially druggable site[36,37]. The K3 data shows that we have isolated a non-covalent KRAS binder and have identified a druggable pocket and pharmacophore combination

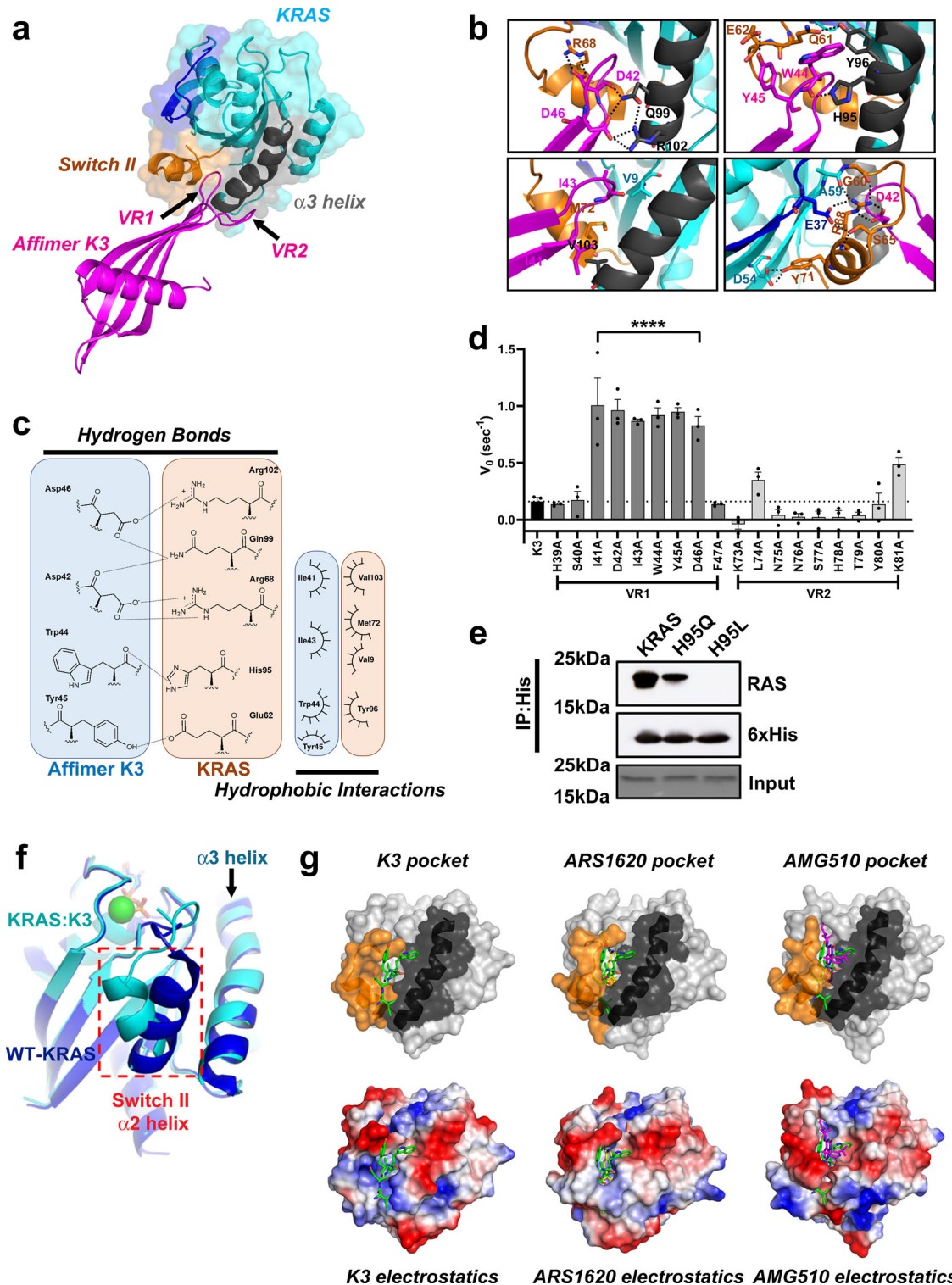

through which to inhibit KRAS preferentially over other RAS isoforms.

**Disruption of the intracellular RAS:Affimer interactions can be used to identify compounds that bind in the pockets.** Having shown Affimers can identify and probe pockets on RAS, we explored the potential of utilizing these interactions as tools to identify compounds that bind in the pockets. To achieve this, a KRAS:Affimer NanoBRET system was developed, initially

demonstrating that both K6 and K3 interact with KRAS, within cells (Fig. 7a and b) and that greater NanoBRET signal was seen with increased Affimer to KRAS ratio, whilst the control Affimer shown no evidence of an interaction. Subsequently, the impacts of small molecule inhibitors that bind in the SI/SII and SII pockets, respectively, on the NanoBRET signal were assessed. Increasing concentrations of the SI/SII-pocket binding compound, BI-2852[11], reduced the NanoBRET signal from the Affimer K6:KRAS interaction. This reduction in signal is compatible with BI-2852 displacing Affimer K6 from KRAS and the NanoBRET

**Fig. 6 Variable region 1 of Affimer K3 explores a druggable pocket between the switch II region and α3 helix of KRAS. a** Affimer K3 (magenta) was co-crystallized with KRAS$^{GDP}$ (cyan) and solved to a resolution of 2.1 Å. The switch I (deep blue), switch II (orange) and α3 helix (dark gray) are depicted, showing their relative positioning around variable region 1 of Affimer K3. **b** Intramolecular and intermolecular interactions in the KRAS:Affimer K3 co-crystal structure is depicted; black dotted lines represent hydrogen bonds. **c** All intermolecular interactions are shown in planar form. Hydrogen bonds are shown as short dotted black lines between the contributing atoms; electrostatic interactions are shown as long dashed black lines additional hydrophobic interactions are represented by arcs, their spokes radiating towards the residues they contact (Data was generated using PDBePISA (CCP4i)[31] and verified in MacPyMOL). **d** Alanine scanning data of the variable regions of Affimer K3 highlights Affimer residues important for inhibition of nucleotide exchange. Unaltered K3 is shown in black, variable region 1 residue are shown in dark gray, and variable region 2 residues in light gray (One-way ANOVA with Dunnett's post hoc test ****$p < 0.0001$, $n = 3$ independent experiments). **e** Mutation of KRAS H95 affects the ability of Affimer K3 to bind, H95Q and H95L represent the residues in HRAS and NRAS, respectively, a representative blot is shown ($n = 3$ independent experiments). **f** Binding of Affimer K3 causes a conformational shift to Switch II compared to WT-KRAS$^{GDP}$. The KRAS molecule (cyan) from KRAS:Affimer K3 co-crystal structure was overlaid with WT-KRAS$^{GDP}$ (deep blue; PDB code: 4OBE). Conformational shifts were observed in the switch II region (red-dotted box). **g** Alterations in the conformation of the Switch II region (orange and α3 helix (black) (top row) and the corresponding alterations in the electrostatics (bottom row). Residues 41–45 of Affimer K3 (green) shown with KRASGDP (left-hand panels), overlaid with the co-crystallized KRAS:ARS1620 structure (middle panels, PBD: 5V9U) and KRAS: AMG510 structure (right-hand panels, PDB: 6OIM) (ARS1620 is shown in yellow and AMG510 is shown in magenta). Data are mean ± SEM, $n = 3$ independent experiments. Images were generated in MacPyMOL v1.7.2.3, and ChemDraw Prime 16.0. VR variable region.

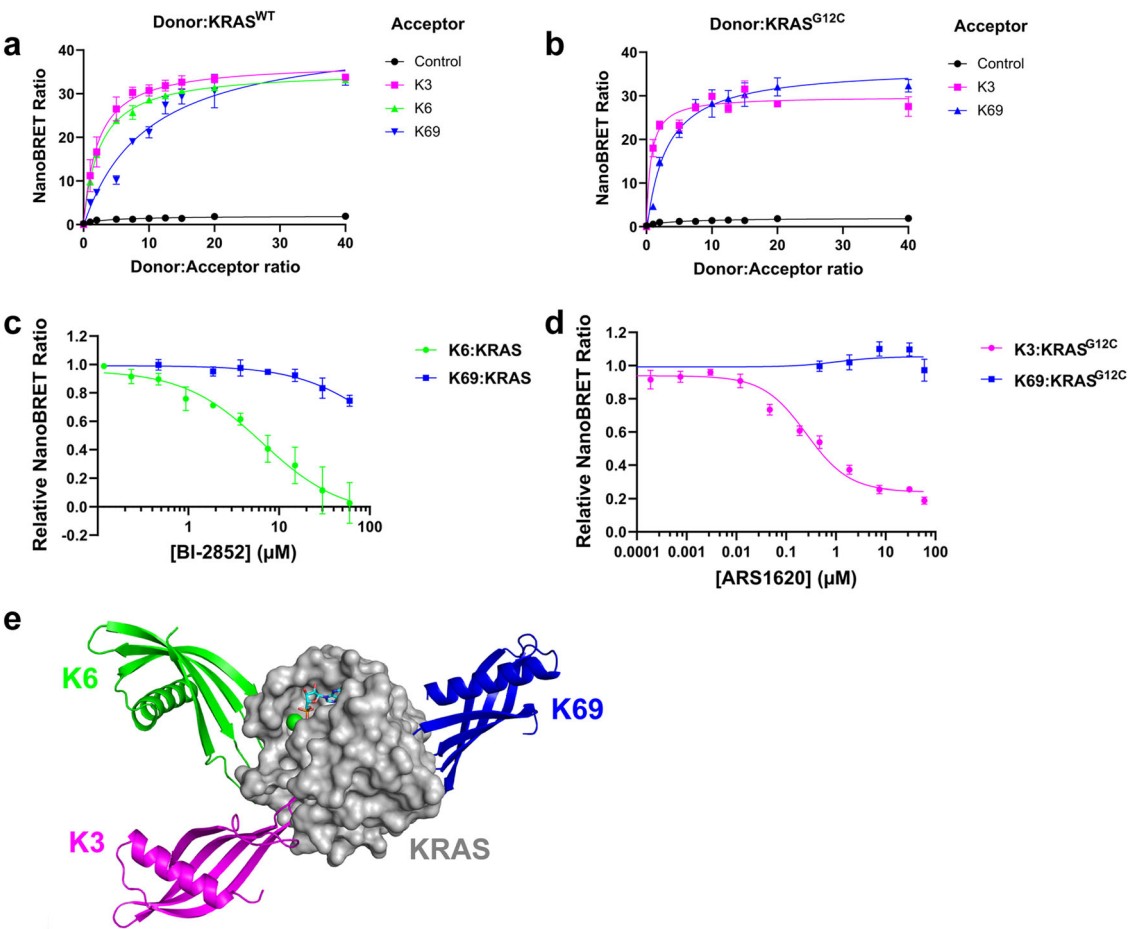

**Fig. 7 Affimer:KRAS NanoBRET can be used to identify small molecules which bind in the SI/SII or SII/α3 pocket.** Increased Affimer (acceptor) to KRAS (donor) ratio increases the NanoBRET signal as measured by NanoBRET Ratio for Affimers K3, K6, and K69 with KRAS$^{WT}$(**a**)) and Affimers K3 and K69 with KRAS$^{G12C}$(**b**)). Small molecule BI-2852 binds in the SI/SII pocket and increasing concentrations displace Affimer K6 reducing the NanoBRET Ratio with no impact on NanoBRET signal from Affimer K69 that binds between helix 4 and helix 5 (**c**)). ARS-1620 covalently tethers to C12 in KRAS$^{G12C}$ and occupies the SII pocket, and increasing concentrations displace Affimer K3 reducing the NanoBRET Ratio with no impact on NanoBRET signal from Affimer K69 (**d**)). **e** Affimer K69 (blue) binds the allosteric lobe between helices 4 and 5 on the opposite side of KRAS (gray) to Affimers K3 (magenta) and K6 (green). Data are mean ± SEM, fitted to 3 parameters [agonist]/[inhibitor] vs. response model in Prism, $n = 3$ independent experiments for all assays. Images were generated in MacPyMOL v1.7.2.3 from PDB codes 6YXW (K3), 6YR8 (K6), and 7NY8 (K69). Control Affimer (Variable regions of AAAA and AAE).

signal was completely abolished with a dose of 60 μM of BI-2852 (Fig. 7c). Similarly increasing concentrations of the SII-pocket binding compound, ARS-1620, reduced the K3:KRAS$^{G12C}$ NanoBRET signal. KRAS$^{G12C}$ was used as ARS-1620 requires a disulfide linkage to C12 to bind and access the SII pocket[12]. Affimer K3:KRAS$^{G12C}$ showed an increased NanoBRET signal with an increased Affimer:KRAS ratio that was comparable to K3:KRAS$^{WT}$ (Fig. 7a and b). ARS-1620 disrupted the K3:KRAS$^{G12C}$ interaction at lower concentrations than BI-2852 did for K6 (0.0468 μM for ARS-1620 vs. 0.468 μM for BI-2852), however complete signal abolition was not achieved with ARS-1620 even at the top dose of 60 μM (Fig. 7d). Neither BI-2852 nor ARS-1620 disrupted the NanoBRET signal from Affimer K69 (Fig. 7c and d) which binds at a distal site to both pockets, between helices 4 and 5 of the allosteric lobe, (Fig. 7e, PDB code 7NY8) and does not inhibit nucleotide exchange (Fig. 1a). Thus, we have demonstrated that the inhibitory RAS-binding Affimers that bound in pockets on RAS can be used as tools to identify compounds that bind in the same pockets. Using this assay, it may be possible to find RAS inhibitors that target KRAS$^{WT}$ in the SII/α3 pocket specifically.

## Discussion

We have isolated RAS-binding Affimer reagents that inhibit RAS both in vitro and in cells. The Affimer proteins generated show nanomolar affinities for KRAS together with IC$_{50}$ values in the nanomolar range for inhibition of SOS1-mediated nucleotide exchange. Furthermore, they are functional intracellularly demonstrating inhibition of the MAPK pathway as assessed by ERK phosphorylation levels. Structural analyses showed that the Affimer proteins interact with RAS within druggable pockets, notably identifying a pocket between the Switch II region and the α3 helix, with a non-covalent binder. Thus, we have exemplified a site for the development of compounds to inhibit both wild-type and G12 mutant forms of KRAS, together with a pharmacophore as a starting point for this approach.

The biochemical and cellular profiles of the Affimer proteins used in this study are comparable with scaffold-based biologics that have previously been identified that also inhibit RAS, again with nanomolar affinities and IC$_{50}$ values[16,17,19–22,38]. However, the majority of these do not distinguish between RAS variants, and/or mutants, and structural analyses reveal that these pan-RAS inhibitors are binding in the Switch I/II region, except for the NS1 monobody that binds the α4-β6-α5 dimerization domain[20], and the DARPins K13 and K19[16] (Supplementary Fig. 2) as discussed below. The binding positions of the scFV, iDab6, and the DARPins K27 and K55 all span the SI/SII pocket[17,22], which is the location of Affimer K6 binding; however, none of these other biologics have been shown to protrude into the pocket. Indeed, structural analysis of the K6:KRAS complex showed that a tripeptide motif, P42, W43, and F44, inserts into this SI/SII pocket in a manner that mimics the binding of known small molecules targeting this pocket. The aromatic indole ring of W43 extends into the pocket, a motif seen with all the compounds targeting this site. The interactions of the compounds are then diverse compared to Affimer K6 (Fig. 4e). It would be interesting to determine if compounds based on the K6 pharmacophore were more potent than the current compounds, as K6 binds with a higher affinity than any published reagent and shows comparable inhibition[5–8,11], and indeed requires micromolar concentrations of the highest affinity inhibitor, BI-2852, to displace it from KRAS.

Thus, Affimer K6 demonstrates that the use of biologics with small interaction interfaces can not only bind and inhibit difficult-to-drug proteins but also identify and probe druggable

pockets on such proteins, potentially acting as templates for small molecules. Importantly, the K3 Affimer shows inhibition of RAS but also demonstrates a preference for KRAS over the HRAS and NRAS variants. To our knowledge, the only other biologics to express such RAS variant specificity are DARPins K13 and K19[16]. This preferential behavior is underpinned by the involvement of the H95 residue unique to KRAS; mutation of this residue abolished binding of both Affimer K3 and the DARPins K13 and K19[16]. This ablation was more complete with mutation to glutamine for the DARPins but was still significant with Affimer K3. These differences may in part be due to the distinct binding locations of the DARPins K13 and K19 on the allosteric lobe side of H95, whereas K3 binds on the effector lobe side and locks KRAS in a conformation where a pocket is revealed[16]. The residues involved in this pocket, specifically Q61, underlie the mutational preferences of Affimer K3 for wild-type/G12 mutants vs. Q61; this selectivity is not seen with DARPins K13 and K19[16]. The specificity of DARPin K19 for KRAS has been exploited as a macro drug fused with an E3 ligase and, whilst proteolysis of both mutant and wild-type KRAS occurred, only cells expressing mutant KRAS were killed both in vitro and in vivo[39]. Thus, this demonstrates the importance of being able to target the KRAS-isoform specifically, but irrespective of its mutant status, for utility as a potential cancer therapy. The interaction between Affimer K3 and KRAS provides an insight into how this may be achieved with small molecules further advancing this area of research.

The pocket revealed in wild-type KRAS by Affimer K3 binding is a previously unseen conformer of the SII pocket[3,4,9,10,12–15], which we have termed the SII/α3 pocket, and it coincides with a cryptic groove identified computationally[4,12]. A similar conformer of the SII pocket has previously been targeted by a cyclic peptide, KRpep-2d, in the KRAS$^{G12D}$ mutant[15]. However, whilst showing nanomolar inhibition in nucleotide exchange assays, KRpep-2d has only shown micromolar efficacy in cells and was deemed not sufficiently efficacious for in vivo studies[14,15]. In contrast, the covalently-tethered KRAS$^{G12C}$ inhibitors, the ARS series, and the most recent iterations are in clinical trials[9,10,12,13] demonstrating the clinical importance of this pocket. Whilst this compound series has yielded the most clinically promising RAS-inhibitors to date, its dependence on covalent tethering to C12 restricts its utility to KRAS$^{G12C}$ mutant cancers only. Affimer K3 binds to an SII-derived pocket non-covalently and demonstrates a similar degree of in vitro potency to AMG510 with IC$_{50}$ values for nucleotide exchange of 0.15 μM and 0.09 μM, respectively[12]. We, therefore, suggest that targeting this region is a strong approach for allosteric inhibition of RAS.

As Gentile et al.[40] noted, for binding to the SII pocket, a substituted phenolic ring is required for insertion within the subpocket formed by V9, R68, D69, and M72, and to form hydrogen bonds with R68 and D69 residues. Affimer K3 fulfills these criteria with the aromatic ring of W44 extending into this subpocket and its surrounding residues, S40 and D42 forming the necessary hydrogen bonds, thus the SII/α3 pocket shares a key subpocket with the SII pocket. Indeed, K3 also forms hydrogen bonds with H95 and Q99 in common with AMG510, albeit with different orientations of H95. Nevertheless, the positioning of the α2 helix of SII is significantly different with K3 inducing a conformation where the α2 helix is distal to the α3 helix, and AMG510 a more closed conformation where the α2 helix is semi-distal to the α3 helix, but the loop region of SII is held across the pocket due to hydrogen bonding between AMG510 and KRAS E63. Whilst ARS-1620, the most potent ARS compound, leaves the helices proximal to one another as seen in WT-KRAS$^{GDP}$ (PDB: 4OBE[32]), (Fig. 6g)[10,12]. These differences suggest that there may be an extended pocket area for small molecules based on the

K3 pharmacophore, as identified by mutational analysis, to exploit. This is supported by the competitive NanoBRET as the highest dose of ARS-1620 could not fully abolish the K3:KRAS NanoBRET signal. Providing small molecules, based on the K3 pharmacophore, engage the H95 that governs the KRAS-specificity of Affimer K3 and the DARPins K13/K19, it may be possible to achieve the first non-covalent small molecule inhibitors of KRAS via the SII/α3 pocket that may have similar properties to the E3-ligase fused DARPin K19 that possess the ability to selectively kill mutant KRAS cells[39]. This is an exciting avenue to be explored with future studies.

Our work presented here demonstrates the concept of using biologics that bind with a relatively small interface as precursors for the development of small-molecule inhibitors for difficult-to-drug proteins. This has the potential to add an alternative strategy for drug discovery to commonly used methods such as computational analysis, covalent tethering that is dependent on suitable residues, and experimental screening approaches. The Affimer proteins identified in this study inhibited RAS, by binding to shallow pockets previously identified, or pockets derived from those previously identified, with comparable affinities and in vitro efficacies to the best small molecules available that target these pockets. This highlights the ability of Affimer proteins to select conformers of target proteins and reveal druggable regions on protein surfaces concurrent with pharmacophore identification. Indeed, it will be interesting to use the pharmacophore motifs identified in this study as templates for novel series of RAS-binding small molecules, and for potential hit-to-lead optimization using the Affimer-RAS NanoBRET system, as has been previously achieved with RAS-binding biologics[5,7]. The approach utilized in this study is likely to be applicable to other important therapeutic targets and provides a useful pipeline for drug discovery.

## Methods

**Protein production**. The human wild-type KRAS (Isoform b), HRAS, and NRAS gene sequences (residues 1–166, with an N-terminal His-tag and C-terminal BAP-tag, were synthesized by GenScript (Piscataway, USA) and cloned into pET11a (all primers used in this study are detailed in Supplementary Table 5). RAS mutants G12D, G12V, Q61H, H95Q, and H95L were produced by Quikchange™ site-directed mutagenesis using the wild type as a template. RAS proteins were produced in BL21 STAR™ (DE3) *E. coli* induced with 0.5 mM IPTG and grown overnight at 20 °C at 150 rpm. Cells were harvested by centrifugation at 4816×*g* for 15 min at 4 °C and resuspended in 20 mM Tris, pH 7.5, 500 mM NaCl, 10 mM Imidazole, 5% Glycerol, supplemented with EDTA-free protease inhibitor, 0.1 mg/ml lysozyme, 1% Triton X-100 and 10U/ml Benzonase nuclease. Proteins were purified from the supernatant by Ni-NTA chromatography and size exclusion chromatography into RAS buffer (20 mM Tris, 100 mM NaCl, 10 mM MgCl2, 1 mM DTT, 5% Glycerol, pH 7.5). GST-thr-RAF1-RBD was provided as a gift from Dominic Esposito (Addgene plasmid # 86033) and produced as previously described[41]. GST-thr-RAF1-RBD supernatants were used for RAS:RAF immuno-precipitation. Human SOS1 catalytic domains (SOS1cat) coding region (residues 564-1059) with an N-terminal His-tag in pET11a were produced in BL21 Star™ DE3 *E. coli* following 0.5 mM IPTG induction and grown overnight at 25 °C at 150 rpm. Cells were harvested by centrifugation at 4816×*g* for 15 min at 4 °C. Cell pellets were lysed in 20 mM Tris-HCl pH 8; 300 mM NaCl; 20 mM imidazole; 5% glycerol supplemented with 1% Triton-X100, EDTA-free protease inhibitor, 0.1 mg/ml lysozyme and 10 U/ml Benzonase nuclease. The lysate was centrifuged at 12,000×*g* for 20 min then applied to Ni-NTA resin. Proteins were eluted using 20 mM Tris-HCl pH 8; 500 mM NaCl; 300 mM Imidazole; 5% Glycerol and dialyzed into 50 mM Tris-HCl pH 7.5; 100 mM NaCl; 1 mM DTT.

**RAS nucleotide loading**. RAS was desalted into nucleotide loading buffer (25 mM Tris-HCl, 50 mM NaCl, 0.5 mM MgCl2 pH 7.5) using a Zeba spin column according to the manufacturer's instructions (ThermoFisher). MANT-GDP (mGDP, SigmaAldrich) or GppNHp (SigmaAldrich) was added in a 20 fold excess over RAS together with DTT and EDTA to a final concentration of 1 mM and 5 mM, respectively, in a volume of 130 μl and incubated at 4 °C for 1 h. MgCl2 was added then in a 9 fold excess over EDTA and incubated for a further 30 min at 4 °C. Loaded RAS was then desalted into nucleotide exchange buffer (20 mM HEPES pH 7.5, 150 mM NaCl, 10 mM MgCl2) using a Zeba spin column. Nucleotide loading was confirmed by native mass spectrometry.

**Phage display and Affimer protein production**. Affimers against GDP-bound KRAS were isolated by phage display[26]. Biotinylated KRAS (1–166)GDP (EZ-Link® NHS-Biotin, Thermo Scientific) was immobilized on blocked (2× blocking buffer, Sigma containing 10 mM MgCl2) streptavidin wells. The Affimer phage library was applied for 2 h and unbound phage removed by PBS-T washes (27 times). Bound phage was eluted in a two-phase step, firstly with 0.2 M glycine pH 2.2 neutralized with 15 ml of 1 M Tris-HCl, pH 9.1 and then 7.18 M triethylamine, pH 11 neutralized with 1 M Tris-HCl, pH 7. Three panning rounds were undertaken and after the final panning around 24 randomly picked colonies were used in phage ELISA with positive clones sent for sequencing[26]. The seven unique sequences were cloned into pET11 using Affimer-His primers (Supplementary Table 5). RAS-binding Affimers were produced in BL21 STAR™ (DE3) *E. coli* (Life Technologies, Invitrogen) and affinity purified using Ni-NTA resin. The cross-reactivity against GppNHp-bound KRAS was determined by ELISA[26].

**Guanine nucleotide exchange assays**. Nucleotide exchange buffer was supplemented with 0.4 mM GTP (SigmaAldrich) and 0.5 μM SOS1cat for experiments involving WT RAS, or 2 μM SOS1cat for experiments involving mutant RAS proteins. The Affimer proteins were then diluted with the GTP-SOS1cat supplemented nucleotide exchange buffer to make 20 μM stock solutions, which were then used for a 2-fold serial dilution series using the supplemented nucleotide exchange buffer. A 1 μM stock of the WT/mutant RASmGDP protein was prepared by diluting the stock RAS in nucleotide exchange buffer supplemented with 2 mM DTT. Solutions were incubated at 37 °C for 10 min before the assay. The reaction was initiated by the addition of Affimer/SOS1cat/GTP solution to RAS/DTT containing solution. Changes in fluorescence were measured by a fluorescence spectrometer (Tecan Spark) in a Corning black, flat-bottomed, non-binding 384 well plate at 440 nm every minute for 90 min. The data were then normalized to initial fluorescence reading and fitted to a single exponential decay using OriginPro 9.7.0 software (OriginLab, Massachusetts). The derived rates were normalized to these of RAS-SOS1 minus RAS-only samples and fitted to the Hill equation ($y$ = START + (END–START) (xn/kn + xn)) from which the IC$_{50}$ values were calculated.

**RAS interaction assays**. The interaction of KRAS with RAF-RBD or Affimer K3 was assessed by immunoprecipitation using a KingFisher Flex (ThermoFisher Scientific). Glutathione magnetic agarose beads were blocked overnight at 4 °C with 2× blocking buffer (Sigma, B6429) then incubated with RAF-RBD-GST supernatants for 1 h at room temperature. Simultaneously 1 μg/μl of KRAS-GppNHp was incubated with 0.6 μg/μl of Affimer or PBS (no Affimer control) in a total volume of 100 μl PBS. Beads were washed 3 times with assay buffer (125 mM Tris, 150 mM NaCl, 5 mM MgCl2, 1 mM DTT, 0.01% Tween-20, pH 8.0) and KRAS:Affimer solutions added and incubated for 1 h at room temperature. Beads were washed 4 times (15 secs/wash) with assay buffer and proteins were eluted with SDS-PAGE sample buffer (200 mM Tris-HCl, 8% SDS, 20% glycerol, 20% mercaptoethanol, 0.1% (w/v) bromophenol blue, pH 7). RAS:Affimer immunoprecipitation was as for RAS:RAF immunoprecipitation except for His Mag Sepharose™ Ni2+-NTA beads (GE healthcare®) were used and incubated with 20 μg of Affimer K3 for 1 h with agitation. After washing 3 times the beads were mixed with 100 μl of KRAS/KRAS(H95Q)/KRAS(H95L) lysate. For elution of K3:KRAS SDS-PAGE sample buffer was supplemented with 500 mM imidazole. Proteins were analyzed by immunoblot with anti-GST-HRP (1:5000, GeneTex, GTX114099) or anti-6X His tag antibody (HRP) (1:10,000, Abcam, ab1187) and KRAS + HRAS + NRAS (1:1000, Abcam, ab206969) antibodies.

**Immunoblotting**. Protein samples were separated on 15% SDS-PAGE and transferred onto nitrocellulose membranes. These were blocked with 5% milk (SigmaAldrich) in TBS-0.1% Tween 20 (TBST), incubated overnight at 4 °C with primary antibody as detailed, washed 3 times with TBST, and incubated for 1 h at room temperature with HRP-conjugated secondary antibody (1:10,000 goat anti-rabbit HRP, Cell Signaling Technology, CST7074S) if required. Following 3× TBS-T washes the membranes were developed with Immunoblot Forte Western HRP (Millipore), according to the manufacturer's instructions, and imaged using an Amersham™ Imager 600 (GE Healthcare). Uncropped images of all membranes used in this manuscript are shown in Supplementary Fig. 3.

**Cell culture**. HEK293, Panc 10.05, and NCI-H460 cells were purchased from ECACC, UK. RAS-expressing mouse embryonic fibroblasts (MEFs) were from William Burgen at Fredrick National Laboratory, Maryland, USA. SW620 cells were from Professor Mark Hull, University of Leeds, UK. HEK293 and MEF cell lines were maintained in Dulbecco's Modified Eagle Medium (SigmaAldrich) supplemented with 10% fetal bovine serum (FBS) (Gibco), and 4 μg/ml blastocidin S (SigmaAldrich)(MEFs expressing KRAS and NRAS isoforms) or 2.5 μg/ml puromycin (SigmaAldrich) (MEFs expressing HRAS). NCI-H460 and SW620 cell lines were maintained in RPMI-1640 media (SigmaAldrich) supplemented with 10% FBS. Panc10.05 cells were maintained in RPMI-1640 supplemented with 15% FBS and 10U insulin (SigmaAldrich). All cells were maintained at 37 °C in CO2 and were mycoplasma-free.

**RAS immunoprecipitation.** $4 \times 10^5$ HEK293 cells/well were plated in 12 well plates and incubated at 37 °C, 5% CO2 for 24 h before transient transfection with plasmids encoding Affimer-His constructs using Lipofectamine 2000 (ThermoFisher), as per the manufacturer's instructions. After 48 h cells were lysed in NP-40 buffer (50 mM Tris, 150 mM NaCl, 1% NP-40 (v/v), 1× Halt™ protease inhibitor cocktail (ThermoFisher), 1× phosphatase inhibitor cocktail 2 (SigmaAldrich), pH 7.5.) and cleared lysates incubated overnight at 4 °C with Ni-NTA resin. After washing, proteins were eluted in SDS sample buffer and analyzed by immunoblotting with the anti-KRAS + HRAS + NRAS or anti-6X His tag-HRP antibody.

**FLAG-ERK pull-down assays.** Cells were plated into 12-well plates ($1 \times 10^5$ cells/well for HEK293 cells, $2 \times 10^5$ cells/well for SW620 and NCI-H460 cells and $4 \times 10^5$ cells/well for Panc10.05 cells) and incubated at 37 °C, 5% CO2 for 24 h before transfection with a 4:1 DNA ratio of pCMV6-Affimer-tGFP and FLAG-ERK plasmids using Lipofectamine 2000 (SW620, HEK293, NCI-H460) or X-tremeGENE 9 (Roche; Panc10.05). After 24 h cells were serum-starved for 1 h, HEK293 cells were then stimulated with 25 ng/ml EGF (Gibco) for 5 min (other cell lines were not stimulated). Cells were washed with ice-cold DPBS then incubated for 10 min on ice with NP-40 lysis buffer. Cleared lysates were incubated overnight at 4 °C with 20 µl anti-FLAG M2 magnetic beads (SigmaAldrich). The beads were washed 3× with TBS before protein elution by incubation at 95 °C for 5 min in SDS sample buffer. Levels of ERK and pERK were then analyzed by immunoblotting with anti-ERK antibody (1:2000, Abcam, ab184699) and phospho-ERK antibody (1:1000, Abcam, ab76299). Densitometry analysis used ImageJ software v.1.52 (NIH, Maryland).

**pERK immunofluorescence assay.** Cells were plated into 96 well plates ($1 \times 10^5$ cells/ml for HEK293 cells, $2$–$8 \times 10^4$ cells/ml for MEFs) and incubated at 37 °C, 5% CO2 for 24 before transfection with pCMV6-Affimer-tGFP plasmids using Lipofectamine 2000 as per the manufacturer's instructions. After a further 24 h cells were serum-starved for 1–18 h before stimulation with EGF (25 ng/ml) for 5 min, rinsed with PBS, and fixed in 4% paraformaldehyde (VWR) for 15 min. Cells were rinsed with PBS and permeabilized with methanol at −20 °C for 10 min, before rinsing with PBS and blocking (1% milk (SigmaAldrich) in PBS) and incubating with anti-pERK antibody (1:150 Cell Signaling Technology 4370) in blocking solution for 1 h at room temperature followed by 3x PBS rinses and incubating with Hoechst 33342 (1 µg/ml Molecular Probes) and anti-rabbit AlexaFluor 546 or 568 (1:1000 Molecular Probes) in blocking solution for 1 h at room temperature. Following a final set of PBS washes, plates were scanned and images collected with an Operetta HTS imaging system (PerkinElmer) or ImageXpress Pico (Molecular Devices) at ×20 magnification. Images were analyzed with Columbus 2.7.1 (PerkinElmer) or MetaExpress 6.7 (Molecular Devices) software.

**Crystallization, data collection, and structure determination.** Purified Affimer proteins were incubated with KRAS lysates overnight at 4 °C and the complexes purified by Ni-NTA affinity chromatography and size exclusion chromatography using HiPrep 16/60 Sephacryl S-100 column (GE Healthcare). Representative chromatographs and SDS-PAGE of the complexes are shown in Supplementary Fig. 4. The complexes were concentrated to 24 mg/ml (KRAS:K3) 12 mg/ml (KRAS:K6) and 11 mg/ml (KRAS:K69) in 10 mM Tris-HCl pH 8.0, containing 50 mM NaCl, 20 mM MgCl2 and 0.5 mM TCEP (K3 and K6), and 20 mM HEPES (pH 8.0), 150 mM NaCl, 5 mM MgCl2, 0.5 mM DTT (K69), respectively. KRAS:K3 crystals were obtained in 2 M $(NH_4)_2SO_4$, 0.2 M K Na tartrate, and 0.1 M tri-sodium citrate pH 5.6 by sitting drop vapor diffusion. Crystals were flash-cooled in a mixture of 75% mother liquor and 25% ethylene glycol. KRAS:K6 crystals were obtained in 0.1 M $C_2H_3NaO_2$ pH 5, 25% w/v PEG 4 K, 0.2 M $(NH_4)_2SO_4$, 5% MPD by hanging-drop vapor diffusion. Crystals were flash-cooled in 30% w/v PEG 4 K, 0.1 M $C_2H_3NaO_2t$, pH 5, 0.2 M $(NH_4)_2SO_4$, 20 mM MgCl2, 5% PEG 400, 5% MPD, 5% ethylene glycol and 5% glycerol. KRAS:K69 crystals were obtained from the Morpheus Screen 0.12 M Alcohols (0.2 M 1,6-Hexanediol, 0.2 M 1-Butanol, 0.2 M 1,2_Propanediol, 0.2 M 2-Propanol, 0.2 M 1,4-Butanediol, 0.2 M 1,3-Propanediol), 0.1 M Buffer System 1 (1.0 M Imidazole, MES monohydrate) pH 6.5, 30% precipitant mix 1 (20% v/v PEG 500 MME, 10% w/v PEG 20000) by sitting drop vapor diffusion. Crystals were flash-cooled in 75% mother liquor and 25% glycerol. X-ray diffraction data for the KRAS:K6 and KRAS:K69 complexes were recorded on beamline I04-1 (Wavelengths 0.9159 Å and 0.9795 Å, respectively) at the Diamond Light Source, with data for KRAS:K3 being recorded on beamline ID30A-1 (Wavelength 0.9660 Å) at the European synchrotron radiation facility, at 100 K. Data collection statistics are reported in Supplementary Table 4. Diffraction data were processed and scaled with the Xia2 suite of programs[42]. The KRAS:Affimer structures were determined by molecular replacement with the KRAS-GDP structure (PDB 4OBE) and an Affimer structure (PDB 4N6T) excluding the variable regions as the initial search models in the program Phaser[43]. Structures were refined using REFMAC5[44], followed by iterative cycles of the manual model building using COOT[45]. Whilst the final model of the KRAS:K6 structure contains all the residues of the variable regions, the electron density maps for residues 75-80 for both the KRAS:K3 complexes in the asymmetric unit cell were highly disordered with incomplete connectivity even when contoured at low sigma level. This is reflected in the final statistics for our refined structure for the KRas:K3 complex

being higher than other deposited structures in the Protein Data Bank of similar resolution as judged by the Rfactor and FreeRfactor values. Crystallization of the KRAS:K3 complex was very difficult and with only small needle clusters grown. However, the dataset from one such crystal was processed that gave diffraction patterns that were multi-lattice and also anisotropic. The data was processed accounting for the anisotropy, and also for comparison without anisotropy. Separately, model building and refinement rounds were undertaken with each dataset but there was little difference in terms of the final refinement statistics. The deposited structure contains two KRAS:K3 complexes. Overall the quality of the electron density for both the KRAS molecules is good including the density for the GDP molecules. However, both Affimer K3 molecules within the crystal lattice have little inter-molecule interactions due to the crystal packing, resulting in the quality of the density being poor with little to no density for many side-chain atoms and some main chain atoms. In addition, the two Affimer K3 molecules are facing each other across a two-fold axis with poor main chain connectivity density for the regions 74–83. Numerous attempts were made to model this region with the best model presented in the deposited structure with residues S77, H78, and T79 included but as poly-alanines. Hence although the final model statistics could be better for the deposited structure, the electron density maps clearly show the structure of the KRAS and Affimer K3 (apart from the 74–83 region) with very clear density for the interacting residues between the molecules. Model validation was conducted using the Molprobity server[46] with Ramachandran statistics of 94.8% in the favored region and 2 outliers for KRAS:K3, 96.8%, and 0 for KRAS:K6 and 97.9%, and 0 KRAS:K69. Molecular graphics were generated using MacPyMOL version 1.7.2.3. Surface area calculations were performed using the PDBePISA[31] protein–protein interaction server. The KRAS:K6, KRAS:K3, and KRAS:K69 structures have been deposited with the PDB codes 6YR8, 6YXW, and 7NY8, respectively.

**Affimer affinity measurements.** Affimer affinities for KRAS were determined by surface plasmon resonance (SPR) using a BIAcore 3000 (GE Healthcare Europe GmbH). Affimer proteins with a C-terminal cysteine residue were biotinylated with biotin-maleimide (SigmaAldrich) as previously described[26] and immobilized onto streptavidin-coated CM5 sensor chips (Biacore). Biacore experiments were performed at 25 °C in HEPES buffer (20 mm HEPES, pH 7.5, 150 mm NaCl, 10 mm $MgCl_2$, 0.1% Tween 20, 0.1% Triton-X100). KRAS (bound to GppNHp or GDP) was injected at 6.25, 12.5, 25, 50, 100, 200, 400, and 800 nM at a flow rate of 5 µl min$^{-1}$, followed by 3 min stabilization and 10 min dissociation. The on-rates and off-rates and $K_D$ parameters were obtained from a global fit to the SPR curves using a 1:1 Langmuir model, using the BIA evaluation software. Quoted $K_D$ values are the mean ± SEM of three replicate measurements.

**Alanine scanning by site-directed mutagenesis.** To assess the importance of each residue in the Affimer variable regions point mutations to encode sequential alanine residues were introduced by Quikchange™ site-directed mutagenesis. Reactions contained 1× KOD polymerase reaction buffer, 0.2 mM dNTP, 2 mM $MgSO_4$, 0.3 µM of forward and reverse primer, 10 ng DNA template, and 1 U KOD polymerase. PCR amplification consisted of 30 cycles of 20 s at 98 °C, 10 s at 68 °C, and 3.5 mins at 70 °C. Samples were digested with Dpn I for 1 h at 37 °C and introduced by transformation into XL1-Blue super-competent cells. DNA was extracted using QIAprep Spin Miniprep Kit as per the manufacturer's instructions and mutagenesis was confirmed by DNA sequence analysis (Genewiz).

**Construction of K6ΔVR2 mutant.** To generate the Affimer K6ΔVR2 mutant, the 9 residues of the K6 VR2 were replaced with AAE. Affimer K6 VR1 and control Affimer VR2 (AAE) were amplified and subjected to splice overlap extension (SOE) PCR. The spliced product was subcloned into pET11a and Affimer K6ΔVR2 produced as described above[26].

**NanoBRET assays.** The NanoBRET proximity-based assay was carried out using the NanoBRET Nano-Glo detection system (Promega) according to the manufacturer's instructions. Briefly, HEK293 cells were co-transfected with Nluc-KRas and HaloTagged-Affimer plasmids using the FuGENE HD transfection reagent (Promega). Forty-eight hours post-transfection, cells were detached and reseeded in white 384-well plates at a density of $1.2 \times 10^4$ cells/ml in HyClone DMEM no-phenol red (GE Life Sciences), complemented with 0.1% FBS. Reseeded cells were incubated with HaloTag NanoBRET 618 ligand (100 nM final concentration) for an additional 16–24 h at 37 °C, NanoBRET signal was determined 5 mins after the addition of the NanoBRET furimazine substrate (10 µM final concentration) using a Tecan Spark Multimode microplate reader ($\lambda_{Em} = 460 \pm 40$ nm, $\lambda_{Em} = 618 \pm 40$ nm; Life Sciences). Raw NanoBRET ratios were calculated as: RawBRET = 618 nm$_{Em}$/460 nm$_{Em}$ × 1000. Corrected NanoBRET ratios (milliBRET units; mBU) were calculated by discounting the donor-only control Raw NanoBRET ratio. For competition analysis using BI-2852 and ARS-1620, the compounds were titrated using a final concentration range of 0–60 µm alongside the HaloTag NanoBRET 618 ligand and incubated for 16–24 h prior to substrate addition and NanoBRET signal measurement. For compound analyses, the following donor-acceptor ratios were used 1:2 for K3 and K6, and 1:10 for K69, and NanoBRET signal expressed as a ratio relative to the DMSO-only control.

**Statistical analysis**. Data were analyzed in Prism v9.1.0 (GraphPad Software). Normality was tested using Shapiro Wilk test. Data presented are mean ± SEM unless otherwise stated.

**Reporting summary**. Further information on research design is available in the Nature Research Reporting Summary linked to this article.

## Data availability

The X-Ray crystal structures generated during and analyzed during this study are available in the PDB repository [https://www.rcsb.org] with the following codes: 6YXW, 6YR8, and 7NY8. Source data are provided with this paper. The Affimer constructs generated during this study are available under a standard Material Transfer Agreement (MTA) from the University of Leeds via the corresponding author (DCT). Source data are provided with this paper.

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

## Acknowledgements

This work was supported by the Wellcome Trust grant number 102174/B/13/Z, Medical Research Council grant number MR/N020952/1, EPSRC grant number EP/L015005/1, and Technology Strategy Board grant number TS/M001199/1.

## Author contributions

D.C.T., M.J.M., T.A.E., A.L.B., and M.J. conceived the experimental plan. K.Z.H., H.L.M., A.R., and A.L.T. contributed equally to the experimental data. S.E.S., B.P., C.T., K.T., A.A.T., M.A., T.T., K.M.F., T.L.A., T.G.G., and C.H.T. conducted experimental work. All authors performed data analysis and critically reviewed and approved the manuscript.

## Competing interests

M.J. works for Avacta Life Sciences who licensed the Affimers from the University of Leeds. M.J., M.J.M., D.C.T., and A.L.T. all own personal shares in Avacta Life Sciences. The remaining authors declare no competing interests.
