## [Peer Review File · Nature Communications]

Reviewers' comments:

Reviewer #1 (Remarks to the Author):

The manuscript by Haza et al. presents innovative work supported by very interesting data, which together potentially have far reaching implications in the interrogation of new drug targeting sites. This is also a timely manuscript given the urgent need for new approaches in drug design and protein engineering to facilitate work on hitherto undruggable protein drug targets. In particular, the presented tools and approach could help to reveal cryptic drug binding sites relating to the intrinsic conformational plasticity of the protein target.

The authors will need to address the following points:

-The manuscript reports affinities and binding kinetics obtained via SPR, but regrettably the relevant data is not shown via appropriate display items showing sensorgrams and fitted data. Such data will need to be added to the manuscript.

-It is difficult to maintain a good overview and keep track of all the different comparative binding data mentioned throughout the manuscript.

Appropriately consolidated tables would be highly recommended in supplementary materials. This would be expected to substantially improve the readability of the manuscript.

- When rationalizing the differences between the potencies displayed by K3, K6 and K37 in all the different types of experimental undertakings involving cell lines transfected with the selected Affimers, the relative expression levels of those Affimers to KRAS are not discussed.

-It would be useful to include a representative chromatogram and corresponding SDS-PAGE analysis of the pooled fractions of at least one of the Affimer:KRAS complexes used for the structural studies.

-The reported refinement model R-factors for the K3-KRAS complex ($R=23.65$ and $R\text{-free}=27.78$) are higher than what one would expect for a crystal structure to 2.1 \AA resolution. Is X-ray data anisotropy at play, or other issues? Please comment and possibly remedy as necessary.

-In Figure 5C negative charges for the acidic side chains are depicted but not for the arginines: Arg102, Arg68. As is Fig 5C places ionic interactions between Asp-Arg side chains under hydrogen-bonded interactions. This is technically correct given the involvement of h-bonds in nearly all electrostatic interactions, however, it would help the reader to also have appropriate labeling of interactions involving charges.

-Please label the MW marker bands in the blots shown in Supplementary Figure 3.

Reviewer #2 (Remarks to the Author):

KRAS targeting therapeutics are unmet clinical need. Use of affimer proteins as probes for discovery a novel cryptic pockets in KRAS oncogenic protein (that is considered undruggable because of lack of obvious binding pocket small molecule could dock, but it is highly allosteric, so a pocket could form upon ligand binding) sounds like a very attractive concept.

However, the two examples Authors investigate here, the S1-S2 region, and the H3/S2 pocket, both are well known pockets for small molecule binding. Moreover, H3/S2 pocket has been published as a target for a cyclic peptide (biologic and noncovalent; Crystal Structure of a Human K-Ras G12D Mutant in Complex with GDP and the Cyclic Inhibitory Peptide KRpep-2d. Sogabe,S et al ACS Med Chem Lett. 2017 Jul 13; 8(7): 732–736. doi: 10.1021/acsmedchemlett.7b00128

The H3 and H95 interactions have been investigated in "KRAS-specific Inhibition Using a DARPin Binding to a Site in the Allosteric Lobe" from Terence Rabbitts' group DOI: 10.1038/s41467-019-10419-2. Rabbitts' group paper highlights tryptophan residue in their DARPin interacting directly with H95.

Unfortunately, this paper lacks a real proof of concept – and that would be finding a novel binding site in KRAS discovered by using affimer probe, that (in the future) could be targeted by a small molecule inhibitors. This would justify the advantage of using affimers over established drug discovery methods, or at least make affimers a competitive tool in the arena of other drug discovery techniques sampling for a cryptic pockets like computational approaches, or experimental like NMR or disulfide tethering. Without this, unfortunately I don't see novelty in this work...

I have several additional comments to the manuscript:

1. Western blot quantification graphs should be included in the Supplementary info. Instead, SPR data as a direct measure of binding (data mentioned in the manuscript) should be included as the figure.
2. In fig 1b (and following figs)– please explain what has been used as a "control"?
3. Fig. 2E is not very convincing, please can you include a detailed explanation or provide better images?
4. Please provide WB images for K, H, and NRAS MEF experiments (Fig 2F). Are levels of expression of these RAS isoforms comparable in these cell lines?
5. Entire results section is not very well written and difficult to follow. One example: why FLAG-ERK transfection was used instead of utilizing endogenous EKR in this experiment?

Reviewers' comments:

Reviewer #1 (Remarks to the Author):

The manuscript by Haza et al. presents innovative work supported by very interesting data, which together potentially have far reaching implications in the interrogation of new drug targeting sites. This is also a timely manuscript given the urgent need for new approaches in drug design and protein engineering to facilitate work on hitherto undruggable protein drug targets. In particular, the presented tools and approach could help to reveal cryptic drug binding sites relating to the intrinsic conformational plasticity of the protein target.

The authors will need to address the following points:

The manuscript reports affinities and binding kinetics obtained via SPR, but regrettably the relevant data is not shown via appropriate display items showing sensorgrams and fitted data. Such data will need to be added to the manuscript.

Sensograms and fitted data have now been added to the manuscript as Figure 5 and subsequent figures renumbered accordingly.

It is difficult to maintain a good overview and keep track of all the different comparative binding data mentioned throughout the manuscript.

Appropriately consolidated tables would be highly recommended in supplementary materials. This would be expected to substantially improve the readability of the manuscript.

We have added Supplementary Table 3 to summarise all the binding and inhibition data discussed in the manuscript.

When rationalizing the differences between the potencies displayed by K3, K6 and K37 in all the different types of experimental undertakings involving cell lines transfected with the selected Affimers, the relative expression levels of those Affimers to KRAS are not discussed.

The results section now includes the following to address this point: *“It is possible that the differences in responses to Affimers between cell-lines is a result of variations in Affimer-expression levels relative to RAS, the impacts of the Affimer proteins on mutant KRAS were therefore also tested using an immunofluorescence assay in conjunction with MEF cells expressing KRAS mutants; G12D, G12V and Q61R allowing only cells with high Affimer-expression where it is probable that RAS is saturated to be analysed.”*

It would be useful to include a representative chromatogram and corresponding SDS-PAGE analysis of the pooled fractions of at least one of the Affimer:KRAS complexes used for the structural studies.

These data have been added as Supplementary Figure 4 and mentioned in the methods section.

-The reported refinement model R-factors for the K3-KRAS complex (R=23.65 and R-free=27.78) are higher than what one would expect for a crystal structure to 2.1 Å resolution. Is X-ray data anisotropy at play, or other issues? Please comment and possibly remedy as necessary.

We concur that the final statistics for our final refined structure for the KRas-K3 complex are higher than other deposited structures in the Protein Data Bank of similar resolution as judged by the Rfactor and FreeRfactor values. Crystallisation of the KRAS-K3 complex was very difficult and we were only able to grow small needle clusters. We were fortuitous to have been able to process a dataset from one such crystal that gave diffraction pattern that were multi-lattice and also anisotropic. The data was processed accounting for the anisotropy, and also for comparison without anisotropy. Separately, model building and refinement rounds were undertaken with each dataset but there was little difference in term of the final refinement statistics.

The deposited structure contains two KRAS-K3 complexes. Overall the quality of the electron density for both the KRAS molecules is good including the density for the GDP molecules. However, both Affimer K3 molecules within the lattice show little inter molecule interactions due to the crystal packing and the quality of the density is not so good with little to no density for many side chains. In addition, the two Affimer K3 molecules are facing each other across a two fold axis with poor main chain connectivity density for the regions 74-83. Numerous attempts were made to model this region with the best model presented in the deposited structure. Hence although the final model statistics could be better for the deposited structure, the electron density maps clearly show the structure of the KRAS and Affimer K3 (apart from the 74-83 region) with very clear density for the interacting residues between the molecules.

-In Figure 5C negative charges for the acidic side chains are depicted but not for the arginines: Arg102, Arg68. As is Fig 5C places ionic interactions between Asp-Arg side chains under hydrogen-bonded interactions. This is technically correct given the involvement of h-bonds in nearly all electrostatic interactions, however, it would help the reader to also have appropriate labeling of interactions involving charges.

We have added positive charges to Arg102 and Arg68 and indicated ionic interactions in Figure 6C (originally 5C).

-Please label the MW marker bands in the blots shown in Supplementary Figure 3.

MW marker bands have now been labeled in Supplementary Figure 3

Reviewer #2 (Remarks to the Author):

KRAS targeting therapeutics are unmet clinical need. Use of affimer proteins as probes for discovery a novel cryptic pockets in KRAS oncogenic protein (that is considered undruggable because of lack of obvious binding pocket small molecule could dock, but it is highly allosteric, so a pocket could form upon ligand binding) sounds like a very attractive concept.

However, the two examples Authors investigate here, the S1-S2 region, and the

H3/S2 pocket, both are well known pockets for small molecule binding. Moreover, H3/S2 pocket has been published as a target for a cyclic peptide (biologic and noncovalent; Crystal Structure of a Human K-Ras G12D Mutant in Complex with GDP and the Cyclic Inhibitory Peptide KRpep-2d. Sogabe,S et al ACS Med Chem Lett. 2017 Jul 13; 8(7): 732–736. doi: 10.1021/acsmchemlett.7b00128

We apologise for missing this paper in the original submission and we have included this peptide throughout the paper as detailed below:

In the introduction: “however, a cyclic peptide, KRpep-2d with preference for G12D mutations has been identified that binds in a similar pocket¹⁴ showing that biologics can also probe pockets in KRAS. It would be interesting to determine whether this pocket can be non-covalently exploited in other RAS isoforms and mutants, giving wider application to RAS-driven cancers.”

In the results: “A similar pocket has previously been reported in the KRAS G12D mutant in complex with a cyclic peptide, KRpep-2d¹⁴. However there are critical differences between the pocket bound to K3 and when bound to KRpep-2d. Notably K3 binding involves the KRAS-specific residue H95 giving rise to the KRAS-specificity seen in our cellular assays, the isoform specificity of KRpep-2d has not been assessed to our knowledge. Additionally K3 binding induces the KRAS intramolecular bonds between Q61 and Y96 without the involvement of residue 12, whereas KRpep-2d requires an aspartic acid residue to induce the same intramolecular bonding network¹⁴.”

In the discussion: “A similar conformer of the SII pocket has previously been targeted by a cyclic peptide, KRpep-2d, in the KRAS G12D mutant¹⁴. However, whilst showing nanomolar inhibition in nucleotide exchange assays, KRpep-2d has only shown micromolar efficacy in cells and was deemed not sufficiently efficacious for in vivo studies^{14,15}. In contrast the covalently-tethered KRAS^{G12C} inhibitors, the ARS series, and the most recent iterations are in clinical trials^{9,10,12,13} demonstrating the clinical importance of this pocket.”

We believe these additions highlighted that whilst the KRpep-2d is similar to Affimer K3 in binding location there are critical differences that give K3 novelty and when compared to KRpep-2d

The H3 and H95 interactions have been investigated in “KRAS-specific Inhibition Using a DARPin Binding to a Site in the Allosteric Lobe” from Terence Rabbitts’ group DOI: 10.1038/s41467-019-10419-2. Rabbitts’ group paper highlights tryptophan residue in their DARPin interacting directly with H95.

This point was discussed in our original manuscript, however the key difference is the DARPins that bind H95 don’t probe the SII/α3 pocket instead sitting on the other side of the α3 helix. This is stated in the discussion: “due to the distinct binding locations of the DARPins K13 and K19 on the allosteric lobe side of H95, whereas K3 binds on the effector lobe side and locks KRAS in a conformation where a pocket is revealed¹⁴.”

We have also added the following to discuss the most recent DARPin paper: “The

specificity of DARPin K19 for KRAS has been exploited as macrodrug fused with an E3 ligase and whilst proteolysis of both mutant and wild-type KRAS occurred only cells expressing mutant KRAS were killed both in vitro and in vivo³⁹. Thus demonstrating the importance of being able to target the KRAS-isoform specifically, but irrespective of its mutant status, for utility as a potential cancer therapy and this is seen with Affimer K3.”

Unfortunately, this paper lacks a real proof of concept – and that would be finding a novel binding site in KRAS discovered by using affimer probe, that (in the future) could be targeted by a small molecule inhibitors. This would justify the advantage of using affimers over established drug discovery methods, or at least make affimers a competitive tool in the arena of other drug discovery techniques sampling for a cryptic pockets like computational approaches, or experimental like NMR or disulfide tethering. Without this, unfortunately I don't see novelty in this work...

We have highlighted the novelty of using Affimer K3 as pharmacophore template to combine both the properties of the mutant-specific KRAS small molecules and the ability to target KRAS specifically with biologics such as DARPin K19 as detailed below: *“These differences suggest that there may be an extended pocket area for small molecules based on the K3 pharmacophore, as identified by mutational analysis, to exploit. Providing such small molecules engage the H95 that governs the KRAS-specificity of Affimer K3 and the DARPins K13/K19 it may be possible to achieve the first non-covalent small molecule inhibitors of KRAS via the SII/α3 pocket that may have similar properties to the E3-ligase fused DARPin K19 that possess the ability to selectively kill mutant KRAS cells³⁹. This is an exciting avenue to be explored with future studies.”*

We have also stated that we see this as useful alternative strategy in drug discovery: *“This has the potential to add an alternative strategy for drug discovery to commonly used methods such as computationally analysis, covalent tethering that is dependent on suitable residues, and experimental screening approaches.”*

Affimer K3 combines the KRAS specificity, as seen with the DARPins, with the ability to open the SII/α3 pocket non-covalently. This is the first time this has been documented demonstrating proof of concept that non-covalent targeting of this pocket is possible.

I have several additional comments to the manuscript:

1. Western blot quantification graphs should be included in the Supplementary info. Instead, SPR data as a direct measure of binding (data mentioned in the manuscript) should be included as the figure.

We have not removed the quantification graph from this figure, but we have added the SPR data to the manuscript as raised by reviewer 1. As the SPR data appears later in the manuscript we feel the quantification graph still warrants a place in Figure 1.

2. In fig 1b (and following figs)– please explain what has been used as a

“control”?

We have clarified what the control Affimer is. The following text has been added at the first mention of the control Affimer in the manuscript and in the figure legends: “*in which the variable regions are AAAA and AAE respectively*”

3. Fig. 2E is not very convincing, please can you include a detailed explanation or provide better images?

We have added arrows to this figure and altered the labeling slightly to make it clear that the GFP-positive cells are Affimer expressing and the ones used for analysis of ERK phosphorylation. The figure legend has been altered to reflect these changes.

4. Please provide WB images for K, H, and NRAS MEF experiments (Fig 2F). Are levels of expression of these RAS isoforms comparable in these cell lines?

Figure 2F is analysis of immunofluorescence images. The figure legend has been altered to clarify this.

We do not know if the expression of the RAS isoforms is comparable, but as the data from these experiments reflects the trends seen with our in-vitro nucleotide exchange we believe this point is not relevant for our data analysis as we focus on cells with high levels of Affimer transfection likely saturating RAS.

5. Entire results section is not very well written and difficult to follow. One example: why FLAG-ERK transfection was used instead of utilizing endogenous EKR in this experiment?

We have address the point of FLAG-ERK transfection in the manuscript with the addition of the following sentence “*A co-transfection approach was used to ensure assessment of ERK1 phosphorylation in Affimer expressing cells only as transfection was not 100% efficient.*” We used this approach as we use a variety of cell lines each with differing transfection efficiencies, and we do assess phosphorylation of endogenous ERK with our immunofluorescence assay.

We do not feel the results section is poorly written as reviewer 1 has not commented on this. If reviewer 2 has any other specific examples where we are lacking clarity we are willing to address these.

REVIEWER COMMENTS

Reviewer #1 (Remarks to the Author):

I remain supportive of this manuscript. Overall, the authors have addressed my remarks to the original submission reasonably well. Nevertheless, the authors have been a bit sloppy in doing so, requiring the following corrections/implementations to improve clarity and data representation:

(1) The SPR data added are very welcome, but the concentration of analyte per sensor gram curve is illegible in the current form of the figure!

(2) The explanation regarding the challenges faced in the crystal structure determination and refinement of the K3-KRAS complex are useful, I would highly recommend to add the entire text of the answer to the Methods section. Furthermore, it might be appropriate to provide a short comment in the main text to alert the reader about the fact that this structure presented with challenges.

(3) Supplementary Table 3: The units corresponding to the IC50 corresponding to Abd-7 (in micromolar) should be specified correctly and not via the colloquial units of μM .

(4) Supplementary Figure 4: Labelling the lanes in the SDS-PAGE gels as a function of fraction number is not useful unless the reader can actually read the fraction numbers in the chromatograms. In the figure provided, fraction numbers are only given in panel (a), alas they are illegible!

Reviewer #3 (Remarks to the Author):

Haza et al describe three Affimers that bind directly to Ras and inhibit SOS mediated nucleotide exchange. Two of these Affimers are described in more detail, K6, which binds to the well-established pocket between SI/SII, and K3, which binds to a novel pocket of SII/ $\alpha 3$. Affimers here exhibit some isoform selectivity and also some for certain mutant alleles of KRAS. Western blot data for binding (IP) and signalling (pERK), as well as cell proliferation assays are employed for a semi-quantitative characterization. Upon that X-ray structural characterization, SPR-binding data and mutational analysis (nucleotide exchange assay) describe the Affimers and their binding mode further. While the findings of biologics that bind with a small surface to Ras and thus explore novel sites is potentially interesting, the manuscript quality is not great.

In general, there is a lack of large enough sample sizes (IP, WB, SPR), sample sizes are not explicitly described, experiments are poorly described and the dose-response relationship can typically not be evaluated in Figs 1-3. The biological effects of Affimers are insufficiently characterized. What other signaling pathways are affected? Altogether the manuscript falls short of its potential.

Major comments:

- 1) Data in Fig 1a,b, and 2 are not quantitative enough, as they are not assessing the dose-dependent effect of the Affimers.
- 2) In general, how can one see/ know what expression level of each Affimer was achieved? This weights even heavier in combination with point 1). Either 1) or 2) needs to be satisfied, better both.
- 3) Quantification of one WB in bar graphs is misleading, as the grey levels on the blots speak for themselves. Instead robust independent biological repeat numbers should be performed and quantified.
- 4) The mode of Affimer delivery to cells is insufficiently described (p. 6 top) and expression is typically not controlled. In the only instance where this is the case (Fig 2a) the legend does not describe the label and detection mode of Affimers.
- 5) It would be important to be able to compare the potency on cell proliferation with the on-target

(e.g. BRET) in cell activity. Such data are completely missing.

6) The reasoning for the co-transfection approach to measure pERK is implausible (Fig 2b,c). I assume endogenous pERK did not sufficiently respond and overexpression of ERK boosted ERK levels.

6) SPR data in Fig 5 are unclear, are these example traces? What was the sample size? How were these experiments performed (immobilization of what etc)?

7) There is some discordance between data in Fig 4c and d or to a lesser extent Fig 6c and d. Can this be more clearly explained? Based on which data were representations in both c panels derived?

8) The reporting and discussion of the structural features of the two crystallized Affimers could be more compact.

Minor comments:

-In Figure 1a sample sizes are only clear when referring to SI Table 2, information in the legend is missing.

-Here and there confusing wording, such as on p8: 'Affimer K3 delineates [differentiates?] between KRAS...

-Structures in Fig 4b and 6b are too small to recognize side chain details.

-SI table 3 lacks clarity, it is difficult to compare numbers with different (or false μM) units and spacing of the columns.

Reviewer #4 (Remarks to the Author):

I very much enjoyed reading this manuscript reporting characterization of RAS-targeted affimers, including structural data. The paper is generally well written and the experiments are high in quality. I generally agree with the authors' interpretation of the data, although I am uncomfortable about the amount of spin when it comes to speculations around how to use these affimers for small molecule discovery. While I see clear potential, the data here fall short of supporting strong claims such as "This work demonstrates the potential of using biologics with small interface surfaces to select novel, druggable conformations" found in the abstract. For me, this sort of claim would have to be supported by data showing that the affimer was used to prospectively discover a new inhibitor, or at the very least to experimentally show that it could have been used to discover things like current RAS compounds (AMG510, etc.). For example, one could do experiments showing competition between AMG510 and the affimer or some such. To be clear, I think the current data is publishable (I do not think the authors need more data), but the language and claims should be toned down in places. Here they are the statements where I feel authors over-reach

"This work demonstrates the potential of using biologics with small interface surfaces to select novel, druggable conformations"

"Our work demonstrates...scaffold-based biologics can act as pharmacophore templates for development of small molecule inhibitors"

"This approach ... exemplifies a novel pipeline for drug discovery."

Also, it is not clear to me in what sense these affimers "identify and probe druggable pockets". Are the authors saying that if an affimer binds to some location, it is likely to be a druggable location? This seems possible, based on the retrospective analysis of known pockets presented, but is it necessarily true as a prospective technique for identification of new binding sites? How can we know which pockets have true potential for meaningful drugability?

REVIEWER COMMENTS

Dear Editors and Reviewers,

We are really pleased with the positive constructive comments and the suggestions for improvements to the manuscript. All the comments have been addressed below and changes to the manuscript have been shown in italics.

Sincerely

Darren C Tomlinson

Reviewer #1 (Remarks to the Author):

I remain supportive of this manuscript. Overall, the authors have addressed my remarks to the original submission reasonably well. Nevertheless, the authors have been a bit sloppy in doing so, requiring the following corrections/implementations to improve clarity and data representation:

We are pleased this reviewer remains very supportive of the manuscript and we have now addressed all the minor comments listed below, which were oversights in the previous submission.

(1) The SPR data added are very welcome, but the concentration of analyte per sensor gram curve is illegible in the current form of the figure!

The graphs have been replotted with enlarged legends.

(2) The explanation regarding the challenges faced in the crystal structure determination and refinement of the K3-KRAS complex are useful, I would highly recommend to add the entire text of the answer to the Methods section. Furthermore, it might be appropriate to provide a short comment in the main text to alert the reader about the fact that this structure presented with challenges.

We have added the text to the methods as suggested and have added the following comment in the main text:

"however crystallisation of this complex was difficult leading to lower than anticipated Rfactor and FreeRfactor values (see Methods for details)"

(3) Supplementary Table 3: The units corresponding to the IC50 corresponding to Abd-7 (in micromolar) should be specified correctly and not via the colloquial units of uM.

This has been altered and the IC50 data is now expressed in nM in line with comment from Reviewer 3.

(4) Supplementary Figure 4: Labelling the lanes in the SDS-PAGE gels as a function of fraction number is not useful unless the reader can actually read the fraction numbers in the

chromatograms. In the figure provided, fraction numbers are only given in panel (a), alas they are illegible!

The chromatographs have been replotted to improve the legibility of the fraction numbers.

Reviewer #3 (Remarks to the Author):

Haza et al describe three Affimers that bind directly to Ras and inhibit SOS mediated nucleotide exchange. Two of these Affimers are described in more detail, K6, which binds to the well-established pocket between S1/S11, and K3, which binds to a novel pocket of S11/alpha3. Affimers here exhibit some isoform selectivity and also some for certain mutant alleles of KRAS. Western blot data for binding (IP) and signalling (pERK), as well as cell proliferation assays are employed for a semi-quantitative characterization. Upon that X-ray structural characterization, SPR-binding data and mutational analysis (nucleotide exchange assay) describe the Affimers and their binding mode further.

While the findings of biologics that bind with a small surface to Ras and thus explore novel sites is potentially interesting, the manuscript quality is not great.

In general, there is a lack of large enough sample sizes (IP, WB, SPR), sample sizes are not explicitly described, experiments are poorly described and the dose-response relationship can typically not be evaluated in Figs 1-3. The biological effects of Affimers are insufficiently characterized. What other signaling pathways are affected? Altogether the manuscript falls short of its potential.

We apologise that the number of repeats were not obvious for the reviewer in the manuscript although all were stated in the figure legends in the original manuscript. As some of the initial experiments were screens, these experiments have only been performed as one biological replicate as is normal for screening. The main point of this manuscript was to demonstrate the ability of the Affimers to identify druggable pockets in a 'hard-to-drug' protein; Ras. Although we agree there is a lot more information to be collected about the effects of the Affimers on downstream signalling, this represents a step beyond this publication and its' current focus. We feel the demonstration that the Affimers inhibit Ras and some downstream effectors is sufficient to show they inhibit Ras function in cells, which represents, for the first time, that a non-covalent inhibitor of this pocket can inhibit cellular Ras function.

Major comments:

1) Data in Fig 1a,b, and 2 are not quantitative enough, as they are not assessing the dose-dependent effect of the Affimers.

We originally did not include this data as the IC50's were stated in the manuscript. However, we agree this data is useful for someone reading the manuscript and we have now included dose-response curves for the 3 Affimers (Fig 1b) we have taken forward from the screen data shown in Fig. 1a. We have clarified the manuscript to make this clear by adding the concentration used for screening (5 µM) has been added together with the following text:

"The dose dependency of nucleotide exchange inhibition by the 3 selected Affimers was then assessed (Fig. 1b)."

We have also shown as GFP intensity increases on a cellular level the inhibition of ERK phosphorylation increases (Fig 2f) and the following text addition to p7 again demonstrating in cell dose dependency.

“We determined if the decrease in pERK nuclear translocation correlated with increased Affimer expression on a cellular level by determining pERK inhibition at a number of different GFP intensities. Increased GFP expression and thus Affimer expression showed a reduction in pERK nuclear translocation with a plateau reached at 1000 arbitrary units (Fig. 2f).”

2) In general, how can one see/ know what expression level of each Affimer was achieved? This weights even heavier in combination with point 1). Either 1) or 2) needs to be satisfied, better both.

We apologise this was not clear within the manuscript and have now clarified that the data presented in Fig 1 is using purified proteins and have included that statement below that the percentage of GFP positive cells did not vary between Affimers in the immunofluorescence assays.

“There were no significant differences ($p=0.429$ One-way ANOVA, in the percentage of tGFP positive cells between the different Affimer constructs.”

3) Quantification of one WB in bar graphs is misleading, as the grey levels on the blots speak for themselves. Instead robust independent biological repeat numbers should be performed and quantified.

As stated above we have clarified the Figure legend to make it clear that the WB now Fig 1d is a representative blot from 3 independent experiments that are quantified in Fig 1c.

“Figure 1: Biochemical analysis of RAS-binding Affimers. a) Nucleotide exchange assay shows 3 Affimers, K3 (green triangles), K6 (turquoise triangles) and K37 (yellow triangles), inhibit SOS1-mediated nucleotide exchange, whilst K19 (orange diamonds) and K68 (magenta triangles) show inhibition of intrinsic nucleotide exchange and K69 (purple hexagons) and K91 (navy stars) do not inhibit nucleotide exchange at 5 μ M. KRAS alone is shown as black squares and in the presence of SOS1^{cat} as red circles. **b)** Affimers K3, K6 and K37 demonstrate dose-response inhibition of KRAS^{WT} nucleotide exchange (Data fitted to the Hill Model ([Affimer] vs response (three parameters)), $n=3$ independent experiments for K3 and K6 and $n=5$ for K37). **c)** Immunoprecipitation of KRAS with GST-RAF-RBD is inhibited by the RAS-binding Affimers, K3, K6 and K37, compared to control Affimer (Variable regions of AAAA and AAE) which does not differ from the no Affimer (PBS). GST alone does not pull-down RAS (Quantification using ImageQuantTL; $n=3$ independent experiments). **d)** A representative Western blot of the pull down experiment from c). Data is mean \pm SEM, ANOVA with Dunnett’s post-hoc test * $p < 0.05$, *** $p < 0.001$, **** $p < 0.0001$. Con. – Control Affimer, RBD-RAS Binding Domain, IP – immunoprecipitation.”

4) The mode of Affimer delivery to cells is insufficiently described (p. 6 top) and expression is typically not controlled. In the only instance where this is the case (Fig 2a) the legend does not describe the label and detection mode of Affimers.

We have clarified that cells were transfected with plasmids for expression of tagged Affimers with the following alteration:

“We then examined whether the Affimer proteins retained their ability to interact with and inhibit RAS in human cells by using HEK293 cells and transiently transfecting in plasmids for expression of His-tagged Affimer proteins.”

The legend for Figure 2 has been adjusted to describe the label and detection mode better.

“Figure 2: Affimers bind to intracellular RAS and inhibit downstream signalling. a) Ni-NTA immunoprecipitation of transiently expressed intracellularly His-tagged Affimers with endogenous RAS from HEK293 cells. RAS-binding Affimers, K3, K6 and K37, pulled down endogenously expressed RAS, whilst the control Affimer did not. A representative blot from 3 independent experiments is shown. **b) and c)** HEK293 cells were co-transfected with FLAG-ERK1 plasmid and pCMV6 encoding tGFP tagged Affimers. Twenty-four hours post transfection, cells were serum-starved and treated with EGF for 5 min. FLAG-ERK1 was precipitated from cell lysates and analysed for phosphorylation. **b)** shows a representative blot from 3 independent experiments quantified in **c)** showing that Affimers K6 and K37 significantly reduced ERK phosphorylation by over 60% while Affimer K3 reduced it by 30%. **d)** RAS-binding Affimers reduce EGF-induced phosphorylation and nuclear translocation of endogenous ERK in HEK293 cells as measured by immunofluorescence as a percentage of the control Affimer, with Affimers K6 and K37 showing inhibition of over 80% whilst Affimer K3 inhibits by 50% in GFP-expressing cells over 1500 arbitrary units (n=3 independent experiments). **e).** Representative images of the effects of RAS-binding Affimers, K3, K6 and K37, and the control Affimer on EGF-stimulated upregulation of pERK in HEK293 cells. A selection of GFP-positive cells (green) expressing RAS Affimers (arrowed) show reduced staining for pERK (yellow). **f)** Assessment of Affimer expression level on pERK inhibition as determined by immunofluorescence, increased GFP expression and thus Affimer expression shows a reduction in pERK nuclear translocation (n=3 independent experiments). **g)** RAS-binding Affimers inhibition of EGF-induced phosphorylation and nuclear translocation of endogenous ERK in mouse embryonic fibroblasts (MEFs) expressing single human RAS isoforms as measured by immunofluorescence as a percentage of the control Affimer. Affimers K6 and K37 shown inhibition in all RAS isoforms, whilst Affimer K3 inhibited KRAS and HRAS to lesser degree with no inhibition of NRAS (n=3 independent experiments). Scale bars are 50µm. Data are mean ± SEM, ANOVA with Dunnett’s post-hoc test * p < 0.05 ** p < 0.01, *** p < 0.001, **** p < 0.0001. Con. – Control Affimer (Variable regions of AAAA and AAE), EGF – epidermal growth factor, tGFP – turbo green fluorescent protein, WCL – whole cell lysate, IP – immunoprecipitation, Empty – transfection reagents only.”

5) It would be important to be able to compare the potency on cell proliferation with the on-target (e.g. BRET) in cell activity. Such data are completely missing.

We agree with the reviewer that assessment of the effect of the Affimers on cell proliferation would be interesting to explore, however we feel this is beyond the scope of this manuscript as the focus was to demonstrate the ability of the Affimers to identify druggable pockets in a ‘hard-to-drug’ protein, Ras. However, we have addressed the issue of on-target cellular activity with the development of an Affimer:RAS NanoBRET system (p16 and Fig 7) which also serves to demonstrate the ability of the RAS-binding Affimers to be used as tools for drug discovery.

“Disruption of the intracellular RAS:Affimer interactions can be used to identify compounds which bind in the pockets

Having shown Affimers can identify and probe pockets on RAS we explored the potential of utilising these interactions as tools to identify compounds that bind in the pockets. To achieve this, a KRAS:Affimer NanoBRET system was developed, initially demonstrating that both K6 and K3 interact with KRAS, within cells (Fig 7a and b) and that greater NanoBRET signal was seen with increased Affimer to KRAS ratio, whilst the control Affimer shown no evidence of interaction. Subsequently the

impacts of small molecule inhibitors that bind in the SI/SII and SII pockets respectively on the NanoBRET signal were assessed. Increasing concentrations of the SI/SII-pocket binding compound, BI-2852,¹¹ reduced the NanoBRET signal from the Affimer K6:KRAS interaction. This reduction in signal is compatible with BI-2852 displacing Affimer K6 from KRAS and the NanoBRET signal was completely abolished with a dose of 60 μ M of BI-2852 (Fig 7c). Similarly increasing concentrations the SII-pocket binding compound, ARS-1620, reduced the K3:KRAS^{G12C} NanoBRET signal. KRAS^{G12C} was used as ARS-1620 requires a disulphide linkage to C12 to bind and access the SII pocket¹². Affimer K3:KRAS^{G12C} showed increased NanoBRET signal with increased Affimer:KRAS ratio that was comparable to K3:KRAS^{WT} (Fig 7 a and b). ARS-1620 disrupted the K3:KRAS^{G12C} interaction at lower concentrations than BI-2852 did for K6 (0.0468 μ M for ARS-1620 vs 0.468 μ M for BI-2852), however complete signal abolition was not achieved with ARS-1620 even at the top dose of 60 μ M (Fig 7d). Neither BI-2852 or ARS-1620 disrupted the NanoBRET signal from Affimer K69 (Fig. 7c and d) which binds at a distal site to both pockets, between helices 4 and 5 of allosteric lobe, (Fig. 7e, PDB code 7NY8) and does not inhibit nucleotide exchange (Fig 1a). Thus, we have demonstrated that the inhibitory RAS-binding Affimers that bound in pockets on RAS can be used as tools to identify compounds that bind in the same pockets. Using this assay, it may be possible to find novel RAS inhibitors especially that target KRAS^{WT} in the SII/ α 3 pocket.”

Figure 7. Affimer:KRAS NanoBRET can be used to identify small molecules which bind in the SI/SII or SII/ α 3 pocket. Increased Affimer (acceptor) to KRAS (donor) ratio increases the NanoBRET signal

as measured by NanoBRET Ratio for Affimers K3, K6 and K69 with KRAS^{WT} (a) and Affimers K3 and K69 with KRAS^{G12C} (b). Small molecule BI-2852 binds in the SI/SII pocket and increasing concentrations displace Affimer K6 reducing the NanoBRET Ratio with no impact on NanoBRET signal from Affimer K69 that binds between helix 4 and helix 5 (c). ARS-1620 covalently-tethers to C12 in KRASG12C and occupies the SII pocket, and increasing concentrations displace Affimer K3 reducing the NanoBRET Ratio with no impact on NanoBRET signal from Affimer K69 (d). e) Affimer K69 (blue) binds the allosteric lobe between helices 4 and 5 on the opposite side of KRAS (grey) to Affimers K3 (magenta) and K6 (green). Data are mean ± SEM, fitted to 3 parameter [agonist]/[inhibitor] vs response model in Prism, n = 3 independent experiments for all assays. Images were generated in MacPyMOL v1.7.2.3 from PDB codes 6YXW (K3), 6YR8 (K6), and 7NY8 (K69). Control Affimer (Variable regions of AAAA and AAE).

6) The reasoning for the co-transfection approach to measure pERK is implausible (Fig 2b,c). I assume endogenous pERK did not sufficiently respond and overexpression of ERK boosted ERK levels.

We agree that the co-transfection approach is not the most ideal method, however, the effects were confirmed by staining endogenous pERK levels (Fig. 2d-f) therefore validating the results observed in Fig 2b and c. Also, as we used a variety of cell lines across the paper that had different transfection efficiencies we used the co-transfection approach to try and negate this variability when assessing the impact of the Affimers. This co-transfection approach has been used by Spencer-Smith et al with the NS1 RAS monobody. This reference is now included in the results section. We have altered the reasoning for using this method, as shown below.

“A co-transfection approach was used to ensure assessment of ERK1 phosphorylation in Affimer expressing cells only, as transfection was not 100% efficient, a similar approach has been previously used²⁰. “

We have assessed effects on endogenous pERK using our immunofluorescence approach as detailed on p6

“To further study the impacts of RAS-binding Affimer proteins on ERK phosphorylation, we developed an immunofluorescence assay to allow the phosphorylation levels and nuclear translocation of endogenous ERK to be examined.”

6) SPR data in Fig 5 are unclear, are these example traces? What was the sample size? How were these experiments performed (immobilization of what etc)?

The graphs have been replotted with enlarged legends and we have clarified the legend to the following:

“Figure 5. Affimers K3 and K6 bind KRAS with nanomolar affinity. SPR measured binding activities for Affimer K6 with GDP bound KRAS (a), GTP bound KRAS (b) and Affimer K3 with GDP bound KRAS (c), GTP bound KRAS (d). Affimers were immobilized on streptavidin-coated CM5 sensor chips via C-terminal biotin and differing concentrations of GDP or GDP bound KRAS flowed over. Representative curves of 3 replicate experiments are shown with experimental data in colour and Langmuir 1:1 fitting curves in black.”

7) There is some discordance between data in Fig 4c and d or to a lesser extent Fig 6c and d. Can this be more clearly explained? Based on which data were representations in both c panels derived?

We disagree that there is discordance between the data. The data are showing different aspects which is an important part of comparing structural data with physical effects. We discuss in the manuscript that residues within K6 form intra-Affimer bonds that stabilise the structure (p9) and Fig 4c only shows the interactions between K6 and KRas and this underlies the discordance with Fig 4d pointed out by the reviewer.

“The importance of these amino acid residues for K6 function was confirmed by mutational analysis. Individual replacement of P42, W43, F44 or Q45 with alanine reduced Affimer-mediated inhibition of nucleotide exchange (Fig. 4d). These data also revealed the importance of residues F40, N47 and R73 for the inhibitory function of Affimer K6. Indeed, complete removal of the second variable region abolished the inhibitory ability of K6 (Fig. 4d ΔVR2). This effect is most likely a result of F40, N47 and R73, and Q45 forming intra-Affimer hydrogen bonds that stabilise the tripeptide, P42, W43, F44 (Fig. 4b – bottom panel).”

We have added the following to the figure legend to highlight this:

“This highlights the residues that are important for both KRAS:K6 interactions and intra-Affimer interactions that stabilize the conformation of Affimer K6.”

The figure legends have been clarified to show that the data used in Fig 4c and Fig 6c “was generated using PDBEPIA³¹ (CCP4i and verified in MacPyMOL).”

8) The reporting and discussion of the structural features of the two crystallized Affimers could be more compact.

We feel the reporting and discussion of the structural features of the Affimers is an important part of the paper and needs a clear description in the manuscript. The whole aim of the manuscript was to demonstrate the Affimer reagents can highlight new structural features of Ras for potential future drug campaigns and therefore the description needs to be full and complete. The other reviewers did not comment and therefore we feel it important to maintain our focus on these aspects of Affimer reagents, however we acknowledge that our description of the K3:KRAS complex was rather fulsome and so we have condensed this section.

Minor comments:

-In Figure 1a sample sizes are only clear when referring to SI Table 2, information in the legend is missing.

The figure legend has altered and now reads

“Figure 1: Biochemical analysis of RAS-binding Affimers. a) Nucleotide exchange assay shows 3 Affimers, K3 (green triangles), K6 (turquoise triangles) and K37 (yellow triangles), inhibit SOS1-mediated nucleotide exchange, whilst K19 (orange diamonds) and K68 (magenta triangles) show inhibition of intrinsic nucleotide exchange and K69 (purple hexagons) and K91 (navy stars) do not inhibit nucleotide exchange at 5 μM. KRAS alone is shown as black squares and in the presence of SOS1^{cat} as red circles. **b)** Affimers K3, K6 and K37 demonstrate dose-response inhibition of KRAS^{WT}

*nucleotide exchange (Data fitted to the Hill Model ([Affimer] vs response (three parameters)), n=3 independent experiments for K3 and K6 and n=5 for K37). c) Immunoprecipitation of KRAS with GST-RAF-RBD is inhibited by the RAS-binding Affimers, K3, K6 and K37, compared to control Affimer (Variable regions of AAAA and AAE) which does not differ from the no Affimer (PBS). GST alone does not pull-down RAS (Quantification using ImageQuantTL; n=3 independent experiments). d) A representative Western blot of the pull down experiment from c). Data is mean ± SEM, ANOVA with Dunnett's post-hoc test * p < 0.05, *** p < 0.001, **** p < 0.0001. Con. – Control Affimer, RBD-RAS Binding Domain, IP – immunoprecipitation."*

-Here and there confusing wording, such as on p8: 'Affimer K3 delineates [differentiates?] between KRAS...

This instance has been changed to "distinguishes"

-Structures in Fig 4b and 6b are too small to recognizes side chain details.

These panels have been enlarged slightly.

-SI table 3 lacks clarity, it is difficult to compare numbers with different (or false uM) units and spacing of the columns.

The data has been converted to nM and column spacing changed.

Reviewer #4 (Remarks to the Author):

I very much enjoyed reading this manuscript reporting characterization of RAS-targeted affimers, including structural data. The paper is generally well written and the experiments are high in quality. I generally agree with the authors' interpretation of the data, although I am uncomfortable about the amount of spin when it comes to speculations around how to use these affimers for small molecule discovery. While I see clear potential, the data here fall short of supporting strong claims such as "This work demonstrates the potential of using biologics with small interface surfaces to select novel, druggable conformations" found in the abstract. For me, this sort of claim would have to be supported by data showing that the affimer was used to prospectively discover a new inhibitor, or at the very least to experimentally show that it could have been used to discover things like current RAS compounds (AMG510, etc.). For example, one could do experiments showing competition between AMG510 and the affimer or some such. To be clear, I think the current data is publishable (I do not think the authors need more data), but the language and claims should be toned down in places.

We thank this reviewer for their supportive comments and we are really pleased this reviewer enjoyed reading the manuscript. We felt it was important to highlight that Affimer reagents isolated in this paper could be used for small molecule discovery and felt this was a major part of the paper. We realise that the reviewer has not asked for more data but as a result we have now developed an Affimer-Ras NanoBRET system to demonstrate their ability as tools for drug discovery. We have now included some competitive NanoBRET to show that the RAS-binding Affimers have the capacity to

act as reagents in drug discovery. We also demonstrate the competition between the RAS compounds and the Affimers further confirming their ability to be used in drug discovery.

“Disruption of the intracellular RAS:Affimer interactions can be used to identify compounds which bind in the pockets

Having shown Affimers can identify and probe pockets on RAS we explored the potential of utilising these interactions as tools to identify compounds that bind in the pockets. To achieve this, a KRAS:Affimer NanoBRET system was developed, initially demonstrating that both K6 and K3 interact with KRAS, within cells (Fig 7a and b) and that greater NanoBRET signal was seen with increased Affimer to KRAS ratio, whilst the control Affimer shown no evidence of interaction. Subsequently the impacts of small molecule inhibitors that bind in the SI/SII and SII pockets respectively on the NanoBRET signal were assessed. Increasing concentrations of the SI/SII-pocket binding compound, BI-2852,¹¹ reduced the NanoBRET signal from the Affimer K6:KRAS interaction. This reduction in signal is compatible with BI-2852 displacing Affimer K6 from KRAS and the NanoBRET signal was completely abolished with a dose of 60 μ M of BI-2852 (Fig 7c). Similarly increasing concentrations the SII-pocket binding compound, ARS-1620, reduced the K3:KRAS^{G12C} NanoBRET signal. KRAS^{G12C} was used as ARS-1620 requires a disulphide linkage to C12 to bind and access the SII pocket¹². Affimer K3:KRAS^{G12C} showed increased NanoBRET signal with increased Affimer:KRAS ratio that was comparable to K3:KRAS^{WT} (Fig 7 a and b). ARS-1620 disrupted the K3:KRAS^{G12C} interaction at lower concentrations than BI-2852 did for K6 (0.0468 μ M for ARS-1620 vs 0.468 μ M for BI-2852), however complete signal abolition was not achieved with ARS-1620 even at the top dose of 60 μ M (Fig 7d). Neither BI-2852 or ARS-1620 disrupted the NanoBRET signal from Affimer K69 (Fig. 7c and d) which binds at a distal site to both pockets, between helices 4 and 5 of allosteric lobe, (Fig. 7e, PDB code 7NY8) and does not inhibit nucleotide exchange (Fig 1a). Thus, we have demonstrated that the inhibitory RAS-binding Affimers that bound in pockets on RAS can be used as tools to identify compounds that bind in the same pockets. Using this assay, it may be possible to find novel RAS inhibitors especially that target KRAS^{WT} in the SII/ α 3 pocket.”

Figure 7. Affimer:KRAS NanoBRET can be used to identify small molecules which bind in the S1/SII or SII/ α 3 pocket. Increased Affimer (acceptor) to KRAS (donor) ratio increases the NanoBRET signal as measured by NanoBRET Ratio for Affimers K3, K6 and K69 with KRAS^{WT} (a) and Affimers K3 and K69 with KRAS^{G12C} (b). Small molecule BI-2852 binds in the S1/SII pocket and increasing concentrations displace Affimer K6 reducing the NanoBRET Ratio with no impact on NanoBRET signal from Affimer K69 that binds between helix 4 and helix 5 (c). ARS-1620 covalently-tethers to C12 in KRAS^{G12C} and occupies the SII pocket, and increasing concentrations displace Affimer K3 reducing the NanoBRET Ratio with no impact on NanoBRET signal from Affimer K69 (d). e) Affimer K69 (blue) binds the allosteric lobe between helices 4 and 5 on the opposite side of KRAS (grey) to Affimers K3 (magenta) and K6 (green). Data are mean \pm SEM, fitted to 3 parameter [agonist]/[inhibitor] vs response model in Prism, n = 3 independent experiments for all assays. Images were generated in MacPyMOL v1.7.2.3 from PDB codes 6YXW (K3), 6YR8 (K6), and 7NY8 (K69). Control Affimer (Variable regions of AAAA and AAE).

Here they are the statements where I feel authors over-reach

We have also addressed each of the following statements has been altered as detailed below:

“This work demonstrates the potential of using biologics with small interface surfaces to select novel, druggable conformations”

“This work highlights the potential of using biologics with small interface surfaces to select novel, druggable conformations”

“Our work demonstrates...scaffold-based biologics can act as pharmacophore templates for development of small molecule inhibitors”

“Our work supports two important concepts in the use of biologics: firstly, they can be used to select for, and stabilise, conformations that are only present as a small fraction of the conformations of the target protein in solution, particularly those that may not be present in extant crystal structures; and secondly, scaffold-based biologics can act as drug discovery tools and/or pharmacophore templates for identification and development of small molecule inhibitors.”

“This approach ... exemplifies a novel pipeline for drug discovery.”

“and presents the exciting potential of a novel pipeline for drug discovery”

Also, it is not clear to me in what sense these affirmers “identify and probe druggable pockets”. Are the authors saying that if an affimer binds to some location, it is likely to be a druggable location? This seems possible, based on the retrospective analysis of known pockets presented, but is it necessarily true as a prospective technique for identification of new binding sites? How can we know which pockets have true potential for meaningful drugability?

By demonstrating the ability of the Affimer reagents to compete with compounds that bind to similar pockets on Ras we now feel it is justified to say the Affimer reagents bind to a druggable pockets. Owing to the differences between the structural data of Affimer K3-Ras and AMG-Ras complexes we hope this could lead to non-covalent inhibitors of this pocket, which have yet to be discovered. The current paper hopefully demonstrates this retrospectively but owing to the fact that Ras has only recently been drugged we feel this, at least, highlights the future potential on other targets. However, the prospective use of Affimer reagents for identifying pockets this will be addressed in future publications.

REVIEWERS' COMMENTS

Reviewer #1 (Remarks to the Author):

The revised manuscript by Haza et al. has addressed my remarks.

However, I noticed that the arginine residues depicted in Figure 6C do not carry formal +1 charges. In fact, I had remarked about this in the context of a previous revision of this manuscript, which led to correction by the authors. Yet for some reason this newly submitted version has regressed to the original issue with that figure!

Reviewer #3 (Remarks to the Author):

The authors have improved their manuscript and largely corresponded to the points raised by this reviewer. They have improved the clarity of data presentation and the description of procedures, in particular they have now updated sample sizes in the figure legends. Notably, data in Figure 7 support their claims of Affimers as probes for pockets relevant in drug discovery.

However, a few minor issues remain:

1) Referring to the second part of their response 6., I still don't understand their reasoning for ERK1 co-transfection. How would that ensure that ERK1 is only assessed in Affimer expressing cells? The distribution of ERK1 and Affimer plasmids and the resulting expression levels will be somewhat stochastic across the cell population. Besides, the sentence structure is not clear. I will leave this at the discretion of the editor to be corrected.

2) Figure 5 legend seems incorrect. They write '...GDP or GDP bound KRAS flowed over...' I suppose this should be 'GDP or GppNHp'.

Daniel Abankwa

Reviewer #4 (Remarks to the Author):

I am satisfied with the revisions.

Dear Editors and Reviewers,

We are really pleased with the positive constructive comments and the suggestions for improvements to the manuscript. All the comments have been addressed below and changes to the manuscript have been shown in italics.

Sincerely

Darren C Tomlinson

Reviewer #1 (Remarks to the Author):

The revised manuscript by Haza et al. has addressed my remarks.

However, I noticed that the arginine residues depicted in Figure 6C do not carry formal +1 charges. In fact, I had remarked about this in the context of a previous revision of this manuscript, which led to correction by the authors. Yet for some reason this newly submitted version has regressed to the original issue with that figure!

The version of Figure 6C submitted includes the +1 charges on the arginine residues. The reversion was an error.

Reviewer #3 (Remarks to the Author):

The authors have improved their manuscript and largely corresponded to the points raised by this reviewer. They have improved the clarity of data presentation and the description of procedures, in particular they have now updated sample sizes in the figure legends. Notably, data in Figure 7 support their claims of Affimers as probes for pockets relevant in drug discovery.

However, a few minor issues remain:

1) Referring to the second part of their response 6., I still don't understand their reasoning for ERK1 co-transfection. How would that ensure that ERK1 is only assessed in Affimer expressing cells? The distribution of ERK1 and Affimer plasmids and the resulting expression levels will be somewhat stochastic across the cell population. Besides, the sentence structure is not clear. I will leave this at the discretion of the editor to be corrected.

We have altered this sentence, it now reads *"A co-transfection approach was used to ensure assessment of ERK1 phosphorylation in Affimer expressing cells only, a similar approach has been previously used²⁰".*

2) Figure 5 legend seems incorrect. They write ...GDP or GDP bound KRAS flowed over... I suppose this should be 'GDP or GppNHp'.

The figure legend has been edited and now reads "GDP or GppNHp bound KRAS flowed over"

Daniel Abankwa

Reviewer #4 (Remarks to the Author):

I am satisfied with the revisions.